# Energy-Constrained Compression for Deep Neural Networks via Weighted Sparse Projection and Layer Input Masking

**Haichuan Yang**[1], **Yuhao Zhu**[1], and **Ji Liu**[1,2]

[1]Department of Computer Science, University of Rochester, Rochester, USA
[2]Kwai AI Lab at Seattle, Seattle, USA
`h.yang@rochester.edu, yzhu@rochester.edu, ji.liu.uwisc@gmail.com`

## Abstract

Deep Neural Networks (DNNs) are increasingly deployed in highly energy-constrained environments such as autonomous drones and wearable devices while at the same time must operate in real-time. Therefore, reducing the energy consumption has become a major design consideration in DNN training. This paper proposes the first end-to-end DNN training framework that provides quantitative energy consumption guarantees via weighted sparse projection and input masking. The key idea is to formulate the DNN training as an optimization problem in which the energy budget imposes a previously unconsidered optimization constraint. We integrate the quantitative DNN energy estimation into the DNN training process to assist the constrained optimization. We prove that an approximate algorithm can be used to efficiently solve the optimization problem. Compared to the best prior energy-saving methods, our framework trains DNNs that provide higher accuracies under same or lower energy budgets.

## 1 Introduction

Deep Neural Networks (DNNs) have become the fundamental building blocks of many emerging application domains such as computer vision (Krizhevsky et al., 2012; Simonyan & Zisserman, 2014), speech recognition (Hinton et al., 2012), and natural language processing (Goldberg, 2016). Many of these applications have to operate in highly energy-constrained environments. For instance, autonomous drones have to continuously perform computer vision tasks (e.g., object detection) without a constant power supply. Designing DNNs that can meet severe energy budgets has increasingly become a major design objective.

The state-of-the-art model compression algorithms adopt *indirect* techniques to restrict the energy consumption, such as pruning (or sparsification) (He et al., 2018; Han et al., 2015a; Liu et al., 2015; Zhou et al., 2016; Li et al., 2016; Wen et al., 2016) and quantization (Gong et al., 2014; Wu et al., 2016; Han et al., 2015a; Courbariaux et al., 2015; Rastegari et al., 2016). These techniques are agnostic to energy consumption; rather they are designed to reduce the amount of computations and the amount of model parameters in a DNN, which do not truly reflect the energy consumption of a DNN. As a result, these indirect approaches only *indirectly* reduce the total energy consumption. Recently, Energy-Aware Pruning (EAP) (Yang et al., 2017) proposes a more direct manner to reduce the energy consumption of DNN inferences by guiding weight pruning using DNN energy estimation, which achieves higher energy savings compared to the indirect techniques.

However, a fundamental limitation of all existing methods is that they do not provide quantitative *energy guarantees*, i.e., ensuring that the energy consumption is below a user-specified energy budget. In this paper, we aspire to answer the following key question: how to design DNN models that *satisfy a given energy budget while maximizing the accuracy?* This work provides a solution to this question through an end-to-end training framework. By end-to-end, we refer to an approach that directly meets the energy budget without relying heuristics such as selectively restoring pruned weights and layer by layer fine-tuning (Han et al., 2015b; Yang et al., 2017). These heuristics are effective in practice but also have many hyper-parameters that must be carefully tuned.

Our learning algorithm directly trains a DNN model that meets a given energy budget while maximiz-

ing model accuracy without incremental hyper-parameter tuning. The key idea is to formulate the DNN training process as an optimization problem in which the energy budget imposes a previously unconsidered optimization constraint. We integrate the quantitative DNN energy estimation into the DNN training process to assist the constrained optimization. In this way, a DNN model, once is trained, by design meets the energy budget while maximizing the accuracy.

Without losing generality, we model the DNN energy consumption after the popular systolic array hardware architecture (Kung, 1982) that is increasingly adopted in today's DNN hardware chips such as Google's Tensor Processing Unit (TPU) (Jouppi et al., 2017), NVidia's Tensor Cores, and ARM's ML Processor. The systolic array architecture embodies key design principles of DNN hardware that is already available in today's consumer devices. We specifically focus on pruning, i.e., controlling the DNN sparsity, as the main energy reduction technique. Overall, the energy model models the DNN inference energy as a function of the sparsity of the layer parameters and the layer input.

Given the DNN energy estimation, we formulate DNN training as an optimization problem that minimizes the accuracy loss under the constraint of a certain energy budget. The key difference between our optimization formulation and the formulation in a conventional DNN training is two-fold. First, our optimization problem considers the energy constraint, which is not present in conventional training. Second, layer inputs are non-trainable parameters in conventional DNN training since they depend on the initial network input. We introduce a new concept, called input mask, that enables the input sparsity to be controlled by a trainable parameter, and thus increases the energy reduction opportunities. This lets us further reduce energy in scenarios with known input data pattern.

We propose an iterative algorithm to solve the above optimization problem. A key step in optimization is the projection operation onto the energy constraint, i.e., finding a model which is closest to the given (dense) model and satisfies the energy constraint. We prove that this projection can be casted into a $0/1$ knapsack problem and show that it can be solved very efficiently. Evaluation results show that our proposed training framework can achieve higher accuracy under the same or lower energy compared to the state-of-the-art energy-saving methods.

In summary, we make the following contributions in this paper:

- To the best of our knowledge, this is the first end-to-end DNN training framework that provides quantitative energy guarantees;
- We propose a quantitative model to estimate the energy consumption of DNN inference on TPU-like hardware. The model can be extended to model other forms of DNN hardware;
- We formulate a new optimization problem for energy-constrained DNN training and present a general optimization algorithm that solves the problem.

## 2 RELATED WORK

**Energy-Agnostic Optimizations**    Most existing DNN optimizations indirectly optimize DNN energy through reducing the model complexity. They are agonistic to the energy consumption, and therefore cannot provide any quantitative energy guarantees.

Pruning, otherwise known as sparsification, is perhaps the most widely used technique to reduce DNN model complexity by reducing computation as well as hardware memory access. It is based on the intuition that DNN model parameters that have low-magnitude have little impact on the final prediction, and thus can be zeroed-out. The classic magnitude-based pruning (Han et al., 2015b) removes weights whose magnitudes are lower than a threshold. Subsequent work guides pruning using special structures (Liu et al., 2015; Zhou et al., 2016; Li et al., 2016; Wen et al., 2016; He et al., 2017), such as removing an entire channel, to better retain accuracy after pruning.

Quantization reduces the number of bits used to encode model parameters, and thus reducing computation energy and data access energy (Gong et al., 2014; Wu et al., 2016; Han et al., 2015a). The extreme case of quantization is using 1-bit to represent model parameters (Courbariaux et al., 2015; Rastegari et al., 2016). Such binary quantization methods are usually trained from scratch instead of quantizing a pre-trained DNN.

**Energy-Aware Optimizations**    Recently, energy-aware pruning (EAP) (Yang et al., 2017) proposes to use a quantitative energy model to guide model pruning. Different from pure magnitude-based pruning methods, EAP selectively prunes the DNN layer that contributes the most to the total energy consumption. It then applies a sequence of fine-tuning techniques to retain model accuracy. The pruning step and fine-tuning step are alternated until the accuracy loss exceeds a given threshold.

Although EAP a promising first-step toward energy-aware optimizations, its key limitation is that it does not provide quantitative energy guarantees because it does not explicitly consider energy budget as a constraint. Our work integrates the energy budget as an optimization constraint in model training.

**Latency-Guaranteed Compression** Lately, model compression research has started providing guarantees in execution (inference) latency, which theoretically could be extended to providing energy guarantees as well. However, these methods are primarily search-based through either reinforcement learning (He et al., 2018) or greedy-search (Yang et al., 2018). They search the sparsity setting for every single layer to meet the given budget. Thus, they may require a large number of trials to achieve a good performance, and may not ensure that the resulting model accuracy is maximized.

## 3 MODELING DNN INFERENCE ENERGY CONSUMPTION

This section introduces the model of estimating energy consumption of a single DNN inference. We consider the widely-used feed-forward DNNs. Note that our proposed methodology can be easily extended to other network architectures as well. In this section, we first provide an overview of our energy modeling methodology (Section 3.1). We then present the detailed per-layer energy modeling (Section 3.2 and Section 3.3), which allow us to then derive the overall DNN energy consumption (Section 3.4). Our energy modeling results are validated against the industry-strength DNN hardware simulator ScaleSim (Samajdar et al., 2018).

DNN model sparsity (via pruning) is well recognized to significantly affect the execution efficiency and thus affect the energy consumption of a DNN model (He et al., 2018; Yang et al., 2017; Han et al., 2015a; Liu et al., 2015; Zhou et al., 2016). We thus use pruning as the mechanism to reduce energy consumption[1]. Note, however, that model sparsity is not the end goal of our paper; rather we focus on reducing the energy consumption directly. Many dedicated DNN hardware chips (a.k.a., Neural Processing Units, NPUs) (Jouppi et al., 2017; Chen et al., 2016; Han et al., 2016; Parashar et al., 2017) have been developed to directly benefit from model sparsity, and are already widely available in today's consumer devices such as Apple iPhoneX, Huawei Mate 10, and Microsoft HoloLens. Our paper focuses on this general class of popular, widely-used DNN chips.

### 3.1 ENERGY MODELING OVERVIEW

A DNN typically consists of a sequence of convolution (CONV) layers and fully connected (FC) layers interleaved with a few other layer types such as Rectified Linear Unit (ReLU) and batch normalization. We focus mainly on modeling the energy consumption of the CONV and FC layers. This is because CONV and FC layers comprise more than 90% of the total execution time during a DNN inference (Chen et al., 2016) and are the major energy consumers (Han et al., 2015a; Yang et al., 2017). Energy consumed by other layer types is insignificant and can be taken away from the energy budget as a constant factor.

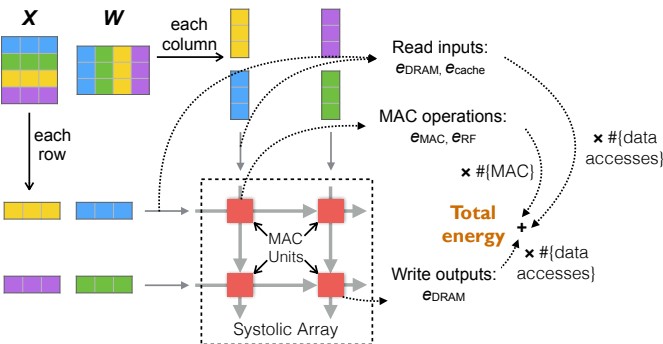

Figure 1: Illustration of the energy cost of computing matrix multiplication $XW$.

A DNN inference's energy consumption is tied to the underlying hardware that performs the inference. In particular, we assume a systolic-array-based DNN hardware architecture. Systolic array (Kung, 1982) has long been know as an effective approach for matrix multiplication. Many DNN hardware architectures adopt the systolic array, most notably the Google Tensor Processing Unit (TPU) (Jouppi

---

[1]Quantization is another useful mechanism to reduce energy consumption. It is orthogonal to the pruning mechanism and they could be combined. This paper specifically focuses on the pruning mechanism.

et al., 2017), Nvidia's Tensor Cores in their most recent Volta GPUs, and ARM's ML Processor. Targeting systolic-array-based DNN hardware ensures that our approach has a wide applicability. However, our modeling and training strategies can generally be applied to other DNN architectures.

Figure 1 shows the overall hardware architecture. The systolic array comprises of several compute units that perform the Multiply-and-Accumulate (MAC) operation, which conducts the following computation: $a \leftarrow a + (b \times c)$, where $b$ and $c$ are the two scalar inputs and $a$ is the scalar intermediate result called "partial sum." MAC operation is the building block for matrix multiplication. The MAC units are organized in a 2-D fashion. The data is fed from the edges, both horizontally and vertically, which then propagate to the MAC units within the same row and columns.

We decompose the energy cost into two parts: computation energy $E_{\text{comp}}$ and data access energy $E_{\text{data}}$. $E_{\text{comp}}$ denotes the energy consumed by computation units, and $E_{\text{data}}$ denotes the energy consumed when accessing data from the hardware memory. Since we mainly use pruning as the energy reduction technique, we now model how $E_{\text{comp}}$ and $E_{\text{data}}$ are affected by DNN sparsity.

## 3.2 Energy Consumption for Computation

CONV layers perform convolution and FC layer perform matrix-vector multiplication. Both operations can be generalized to matrix-matrix multiplication, which involves only the MAC operation (Chetlur et al., 2014; Jouppi et al., 2017). Figure 1 illustrates how a matrix-matrix multiplication is carried out on the systolic array hardware. Given $X$ and $W$, the systolic array computes $XW$ by passing each row of $X$ to each row in the systolic array and passing each column of $W$ to each column in the systolic array. If the width of the systolic array, denoted by $s_w$, is less than the width of $W$, the hardware will fold $W$ column-wise in strides of $s_w$. Similarly, if the height of $X$ is greater than the height of the systolic array ($s_h$), $X$ is folded row-size in strides of $s_h$. Figure 1 illustrates a $2 \times 2$ systolic array multiplying two $4 \times 4$ matrices. Both matrices are folded twice in strides of 2.

Critically, if either inputs of a MAC operation is zero, we can skip the MAC operation entirely and thus save the computation energy. At a high-level, the total computation energy, $E_{\text{comp}}$, can be modeled as $e_{\text{MAC}} N_{\text{MAC}}$, where $e_{\text{MAC}}$ denotes the energy consumption of one MAC operation whereas $N_{\text{MAC}}$ denotes the total number of MAC operations that are actually performed. The challenge is to identify $N_{\text{MAC}}$ for CONV and FC layers, which we discuss below.

**Fully connected layer** Let $X^{(v)} \in \mathbb{R}^{1 \times c}$ be the input vector and $W^{(v)} \in \mathbb{R}^{c \times d}$ be the weight matrix of the FC layer $v$. The FC layer performs matrix-vector multiplication $X^{(v)} W^{(v)}$. The number of MAC operations $N_{\text{MAC}}$ is sum(supp($X$)supp($W$)), where supp($T$) returns a binary tensor indicating the nonzero positions of tensor $T$. So the computation energy for a fully connected layer $v$:

$$E_{\text{comp}}^{(v)} = e_{\text{MAC}}\text{sum}(\text{supp}(X^{(v)})\text{supp}(W^{(v)})) \leq e_{\text{MAC}}\|W^{(v)}\|_0, \tag{1}$$

where the equality is reached when the input is dense.

**Convolution layer** The CONV layer performs the convolution operation between a 4-D weight (also referred to as kernel or filter) tensor and a 3-D input tensor. Let $W^{(u)} \in \mathbb{R}^{d \times c \times r \times r}$ be the weight tensor, where $d$, $c$, and $r$ are tensor dimension parameters. Let $X^{(u)} \in \mathbb{R}^{c \times h \times w}$ be the input tensor, where $h$ and $w$ are the input height and width. The convolution operation in the CONV layer $u$ generates a 3-dimensional tensor:

$$(X^{(u)} * W^{(u)})_{j,y,x} = \sum_{i=1}^{c} \sum_{r',r''=0}^{r-1} X_{i,y+r',x+r''}^{(u)} W_{j,i,r',r''}^{(u)}, \tag{2}$$

where $x, y$ indicate the position of the output tensor, which has height $h' = \lfloor (h + 2p - r)/s \rfloor + 1$ and width $w' = \lfloor (w + 2p - r)/s \rfloor + 1$ ($p$ is the convolution padding and $s$ is the convolution stride).

Tensor convolution (2) can be seen as a special matrix-matrix multiplication (Chellapilla et al., 2006; Chetlur et al., 2014). Specifically, we would unfold the tensor $X^{(u)}$ to a matrix $\bar{X}^{(u)} \in \mathbb{R}^{h'w' \times cr^2}$, and unfold the tensor $W^{(u)}$ to a matrix $\bar{W}^{(u)} \in \mathbb{R}^{cr^2 \times d}$. $\bar{X}^{(u)}$ and $\bar{W}^{(u)}$ are then multiplied together in the systolic array to compute the equivalent convolution result between $X^{(u)}$ and $W^{(u)}$.

Nonzero elements in $X^{(u)}$ and $W^{(u)}$ incur actual MAC operations. Thus, $N_{\text{MAC}} = \text{sum}(\text{supp}(X^{(u)}) * \text{supp}(W^{(u)})) \leq h'w'\|W^{(u)}\|_0$ (the equality means the input is dense), resulting in the following computation energy of a CONV layer $u$:

$$E_{\text{comp}}^{(u)} = e_{\text{MAC}}\text{sum}(\text{supp}(X^{(u)}) * \text{supp}(W^{(u)})) \leq e_{\text{MAC}}h'w'\|W^{(u)}\|_0. \tag{3}$$

### 3.3 Energy Consumption for Data Access

Accessing data happens in every layer. The challenge in modeling the data access energy is that modern hardware is equipped with a multi-level memory hierarchy in order to improve speed and save energy (Hennessy & Patterson, 2011). Specifically, the data is originally stored in a large memory, which is slow and energy-hungry. When the data is needed to perform certain computation, the hardware will load it from the large memory into a smaller memory that is faster and consume less energy. If the data is reused often, it will mostly live in the small memory. Thus, such a multi-level memory hierarchy saves overall energy and improves overall speed by exploiting data reuse.

Without losing generality, we model a common, three-level memory hierarchy composed of a Dynamic Random Access Memory (DRAM), a Cache, and a Register File (RF). The cache is split into two halves: one for holding $X$ (i.e., the feature map in a CONV layer and the feature vector in a FC layer) and the other for holding $W$ (i.e., the convolution kernel in a CONV layer and the weight matrix in an FC layer). This is by far the most common memory hierarchy in DNN hardware such as Google's TPU (Jouppi et al., 2017; Chen et al., 2016; Zhu et al., 2018; Han et al., 2016). Data is always loaded from DRAM into cache, and then from cache to RFs.

In many today's DNN hardwares, the activations and weights are compressed in the dense form, and thus only non-zero values will be accessed. This is done in prior works (Chen et al., 2016; Parashar et al., 2017). Therefore, if the value of the data that is being loaded is zero, the hardware can skip the data access and thereby save energy. There is a negligible amount of overhead to "unpack" and "pack" compressed data, which we simply take away from the energy budget as a constant factor. This is also the same modeling assumption used by Energy-Aware Pruning (Yang et al., 2017).

To compute $E_{\text{data}}$, we must calculate the number of data accesses at each memory level, i.e., $N_{\text{DRAM}}, N_{\text{cache}}, N_{\text{RF}}$. Let the unit energy costs of different memory hierarchies be $e_{\text{DRAM}}, e_{\text{cache}}$, and $e_{\text{RF}}$, respectively, the total data access energy consumption $E_{\text{data}}$ will be $e_{\text{DRAM}} N_{\text{DRAM}} + e_{\text{cache}} N_{\text{cache}} + e_{\text{RF}} N_{\text{RF}}$. We count the number of data accesses for both the weights and input, then combine them together. The detailed derivation of data access energy is included in the Appendix.

### 3.4 The Overall Energy Estimation Formulation

Let $U$ and $V$ be the sets of convolutional layers and fully connected layers in a DNN respectively. The superscript $^{(u)}$ and $^{(v)}$ indicate the energy consumption of layer $u \in U$ and $v \in V$, respectively. Then the overall energy consumption of a DNN inference can be modeled by

$$E(X, W) := \sum_{u \in U} (E_{\text{comp}}^{(u)} + E_{\text{data}}^{(u)}) + \sum_{v \in V} (E_{\text{comp}}^{(v)} + E_{\text{data}}^{(v)}), \tag{4}$$

where $X$ stacks input vectors/tensors at all layers and $W$ stacks weight matrices/tensors at all layers.

## 4 Energy-Constrained DNN Model

Given the energy model presented in Section 3, we propose a new energy-constrained DNN model that bounds the energy consumption of a DNN's inference. Different from prior work on model pruning in which energy reduction is a byproduct of model sparsity, our goal is to directly bound the energy consumption of a DNN while sparsity is just used as a means to reduce energy.

This section formulates training an energy-constrained DNN as an optimization problem. We first formulate the optimization constraint by introducing a trainable mask variable into the energy modeling to enforce layer input sparsity. We then define a new loss function by introducing the knowledge distillation regularizer that helps improve training convergence and reduce overfitting.

**Controlling Input Sparsity Using Input Mask** The objective of training an energy-constrained DNN is to minimize the accuracy loss while ensuring that the DNN inference energy is below a given budget, $E_{\text{budget}}$. Since the total energy consumption is a function of $\|X^{(u)}\|_0$ and $\|W^{(u)}\|_0$, it is natural to think that the trainable parameters are $X$ and $W$. In reality, however, $X$ depends on the input to the DNN (e.g., an input image to an object recognition DNN), and thus is unknown during training time. Therefore, in conventional DNN training frameworks $X$ is never trainable.

To include the sparsity of $X$ in our training framework, we introduce a trainable binary mask $M$ that is of the same shape of $X$, and is multiplied with $X$ before $X$ is fed into CONV or FC layers, or equivalently, at the end of the previous layer. For example, if the input to a standard CONV layer is $X^{(u)}$, the input would now be $X^{(u)} \odot M^{(u)}$, where $\odot$ denotes the element-wise multiplication. In practice, we do not really do this multiplication but only read $X^{(u)}$ on the nonzero positions of $M^{(u)}$.

---

**Algorithm 1:** Energy-Constrained DNN Training.

---

**Input:** Energy budget $E_{\text{budget}}$, learning rates $\eta_1, \eta_2$, mask sparsity decay step $\Delta q$.
**Result:** DNN weights $W^*$, input mask $M^*$.

1 Initialize $W = W_{\text{dense}}, M = \mathbf{1}, q = \|M\|_0 - \Delta q$;
2 **while** *True* **do**
    // Update DNN weights
3     **while** *W has not converged* **do**
4         $W = W - \eta_1 \hat{\nabla}_W \bar{\mathcal{L}}(M, W)$ ;                       // SGD step
5         $W = \mathrm{P}_{\Omega(E_{\text{budget}})}(W)$ ; // Energy constraint projection for weights $W$
6     **end**
7     If previous_accuracy > current_accuracy, exit loop with previous $W$ and $M$;
    // Update input mask
8     **while** *M has not converged* **do**
9         $M = M - \eta_2 \hat{\nabla}_M \bar{\mathcal{L}}(M, W)$ ;                       // SGD step
10         Clamp values of $M$ into $[0, 1]$: assign 1 (or 0) to the values if they exceeds 1 (or negative);
11         $M = \mathrm{P}_{\|M\|_0 \leq q}(M)$ ;   // $L_0$ constraint projection for input mask $M$
12     **end**
13     Round values of $M$ into $\{0, 1\}$;
14     Decay the sparsity constraint $q = q - \Delta q$;
15 **end**
16 $W^* = W, M^* = M$.

---

With the trainable mask $M$, we can ensure that $\|X^{(u)} \odot M^{(u)}\|_0 \leq \|M^{(u)}\|_0$, and thereby bound the sparsity of the input at training time. In this way, the optimization constraint during training becomes $E(M, W) \leq E_{\text{budget}}$, where $E(M, W)$ denotes the total DNN inference energy consumption, which is a function of $X$ and $W$ (as shown in Equation (4)), and thus a function of $M$ and $W$.

**Knowledge Distillation as a Regularizer**   Directly optimizing over the constraint would likely lead to a local optimum because the energy model is highly non-convex. Recent works (Mishra & Marr, 2017; Tschannen et al., 2017; Zhuang et al., 2018) notice that knowledge distillation is helpful in training compact DNN models. To improve the training performance, we apply the knowledge distillation loss (Ba & Caruana, 2014) as a regularization to the conventional loss function. Intuitively, the regularization uses a pre-trained dense model to guide the training of a sparse model. Specifically, our regularized loss function is:

$$\bar{\mathcal{L}}_{\lambda, W_{\text{dense}}}(M, W) := (1 - \lambda)\mathcal{L}(M, W) + \lambda \mathbb{E}_X[\|\phi(X; W) - \phi(X; W_{\text{dense}})\|^2 / |\phi(\cdot; W)|], \quad (5)$$

where $W_{\text{dense}}$ is the original dense model, and $\mathcal{L}(M, W)$ is the original loss, e.g., cross-entropy loss for classification task. $\phi(X; W)$ is the network's output (we use the output before the last activation layer as in Ba & Caruana (2014)), $|\phi(\cdot; W)|$ is the network output dimensionality and $0 \leq \lambda \leq 1$ is a hyper parameter similar to other standard regularizations.

Thus, training an energy-constrained DNN model is formulated as an optimization problem:

$$\min_{M, W} \quad \bar{\mathcal{L}}_{\lambda, W_{\text{dense}}}(M, W) \quad \text{s.t.} \quad E(M, W) \leq E_{\text{budget}}. \quad (6)$$

## 5   OPTIMIZATION

This section introduces an algorithm to solve the optimization problem formulated in (6). The overall algorithm is shown in Algorithm 1. Specifically, the algorithm includes three key parts:

- Initialization by training a dense model. That is,

$$W_{\text{dense}} := \arg\min_W \mathcal{L}(M, W) \quad (7)$$

- Fix $M$ and optimize $W$ via approximately solving (using $W_{\text{dense}}$ initialization):

$$\min_W \bar{\mathcal{L}}(M, W) \quad \text{s.t.} \quad E(M, W) \leq E_{\text{budget}} \quad (8)$$

- Fix $W$ and optimize $M$ by approximately solving :

$$\min_M \bar{\mathcal{L}}(M, W) \quad \text{s.t.} \quad \|M\|_0 \leq q, M \in [\mathbf{0}, \mathbf{1}] \quad (9)$$

After the initialization step (Line 1 in Algorithm 1), the training algorithm iteratively alternates between the second (Line 3-6 in Algorithm 1) and the third step (Line 8-13 in Algorithm 1) while gradually reducing the sparsity constraint $q$ (Line 14 in Algorithm 1) until the training accuracy converges. Note that Equation (7) is the classic DNN training process, and solving Equation (9) involves only the well-known $L_0$ norm projection $P_{\|M\|_0 \leq q}(Q) := \arg\min_{\|M\|_0 \leq q} \|M - Q\|^2$. We thus focus on how Equation (8) is solved.

**Optimizing Weight Matrix** $W$ To solve (8), one can use either projected gradient descent or projected stochastic gradient descent. The key difficulty in optimization lies on the projection step

$$P_{\Omega(E_{\text{budget}})}(Z) := \arg\min_{W \in \Omega(E_{\text{budget}})} \|W - Z\|^2 \tag{10}$$

where $Z$ could be $W - \eta \nabla_W \bar{\mathcal{L}}(W, M)$ or replacing $\nabla_W \bar{\mathcal{L}}(W, M)$ by a stochastic gradient $\hat{\nabla}_W \bar{\mathcal{L}}(W, M)$. To solve the projection step, let us take a closer look at the constraint Equation (4). We rearrange the energy constraint $\Omega(E_{\text{budget}})$ into the following form with respect to $W$:

$$\left\{ W \mid \sum_{u \in U \cup V} \alpha_1^{(u)} \min(k, \|W^{(u)}\|_0) + \alpha_2^{(u)} \max(0, \|W^{(u)}\|_0 - k) + \alpha_3^{(u)} \|W^{(u)}\|_0 + \alpha_4^{(u)} \leq E_{\text{budget}} \right\}, \tag{11}$$

where $W$ stacks all the variable $\{W^{(u)}\}_{u \in U \cup V}$, and $\alpha_1^{(u)}, \alpha_2^{(u)}, \alpha_3^{(u)}, \alpha_4^{(u)}$ and $k$ are properly defined nonnegative constants. Note that $\alpha_1^{(u)} \leq \alpha_2^{(u)}$ and $k$ is a positive integer. Theorem 1 casts the energy-constrained projection problem to a $0/1$ knapsack problem. The proof is included in the Appendix.

**Theorem 1.** *The projection problem in* (10) *is equivalent to the following* $0/1$ *knapsack problem:*

$$\max_{\xi \text{ is binary}} \langle Z \odot Z, \xi \rangle, \quad \text{s.t.} \quad \langle A, \xi \rangle \leq E_{\text{budget}} - \sum_{u \in U \cup V} \alpha_4^{(u)}, \tag{12}$$

*where $Z$ stacks all the variables $\{Z^{(u)}\}_{u \in U \cup V}$, $A$ and $\xi$ are of the same shape as $Z$, and the $j$-th element of $A^{(u)}$ for any $u \in U \cup V$ is defined by*

$$A_j^{(u)} = \begin{cases} \alpha_1^{(u)} + \alpha_3^{(u)}, & \text{if } Z_j^{(u)} \text{ is among the top } k \text{ elements of } Z^{(u)} \text{ in term of magnitude;} \\ \alpha_2^{(u)} + \alpha_3^{(u)}, & \text{otherwise.} \end{cases} \tag{13}$$

*The optimal solution of* (10) *is $Z \odot \xi^*$, where $\xi^*$ is the optimal solution to the knapsack problem* (12).

The knapsack problem is NP hard. But it is possible to find approximate solution efficiently. There exists an approximate algorithm (Chan, 2018) that can find an $\epsilon-$accurate solution in $O(n \log(1/\epsilon) + \epsilon^{-2.4})$ computational complexity. However, due to some special structure in our problem, there exists an algorithm that can find an an $\epsilon-$accurate solution much faster.

In the Appendix, we show that an $(1 + \epsilon)$-approximate solution of problem (10) can be obtained in $\tilde{O}(n + \frac{1}{\epsilon^2})$ time complexity ($\tilde{O}$ omits logarithm), though the implementation of the algorithm is complicated. Here we propose an efficient approximate algorithm based on the "profit density." The profit density of item $j$ is defined as $Z_j^2/A_j$. We sort all items based on the "profit density" and iteratively select a group of largest items until the constraint boundary is reached. The detailed algorithm description is shown in the Appendix (Algorithm 2). This greedy approximation algorithm also admits nice property as shown in Theorem 2.

**Theorem 2.** *For the projection problem* (10), *the approximate solution $W'' \in \Omega(E_{\text{budget}})$ to the greedy approximation algorithm admits*

$$\|W'' - Z\|^2 \leq \|P_{\Omega(E_{\text{budget}})}(Z) - Z\|^2 + \text{Top}_{\|W''\|_0 + 1}((Z \odot Z) \oslash A) \cdot \min((\max(A) - \gcd(A)), R(W'')) \tag{14}$$

*where $\max(A)$ is the maximal element of $A$, which is a nonnegative matrix defined in* (13); $\text{Top}_k(\cdot)$ *returns the $k$-th largest element of $\cdot$; $\oslash$ denotes the element-wise division. $\gcd(\cdot)$ is the largest positive rational number that divides every argument, e.g., $\gcd(0, 1/3, 2/3) = 1/3$. In* (14), $\gcd(A)$ *denotes the greatest common divisor of all elements in $A^2$, and $R(W'')$ denotes the remaining budget*

$$R(W'') = \left( E_{\text{budget}} - \sum_{u \in U \cup V} \alpha_4^{(u)} - \langle A, \text{supp}(W'') \rangle \right).$$

---

[2]Here we assume $A$ only contains rational numbers since gcd is used.

The formal proof is in the Appendix. $W''$ is the optimal projection solution to (10) if either of the following conditions holds:

1. (**The remaining budget is** $0$.) It means that the greedy Algorithm 2 runs out of budget;
2. (**The matrix** $A$ **satisfies** $\max(A) = \gcd(A)$.) It implies that all elements in $A$ have the same value. In other words, the weights for all items are identical.

## 6 EVALUATION

The evaluations are performed on ImageNet (Deng et al., 2009), MNIST, and MS-Celeb-1M (Guo et al., 2016) datasets. For the MS-Celeb-1M, we follow the baseline setting reported in the original paper (Guo et al., 2016), which selects 500 people who have the most face images. We randomly sample 20% images as the validation set. We use both classic DNNs, including AlexNet (Krizhevsky et al., 2012) and LeNet-5 (LeCun et al., 1998), as well as recently proposed SqueezeNet (Iandola et al., 2016) and MobileNetV2 (Sandler et al., 2018).

We compare our method mainly with five state-of-art pruning methods: magnitude-based pruning (MP) (Han et al., 2015b;a), structured sparsity learning (SSL) (Wen et al., 2016), structured bayesian pruning (SBP) (Neklyudov et al., 2017), bayesian compression (BC) (Louizos et al., 2017) and energy-aware pruning (EAP) (Yang et al., 2017). Filter pruning methods (Li et al., 2016; He et al., 2017) require a sparsity ratio to be set for each layer, and these sparsity hyper-parameters will determine the energy cost of the DNN. Considering manually setting all these hyper-parameters in energy-constrained compression is not trivial, we directly compare against NetAdapt (Yang et al., 2018) which automatically searches such sparsity ratios and use filter pruning to compress DNN models. We implement an energy-constrained version of NetAdapt, which is originally designed to restrict the inference latency. Note that MobileNetv2 and SqueezeNet have special structures (e.g. residual block) that are not fully supported by NetAdapt. Thus, we show the results of NetAdapt only for AlexNet and LeNet-5.

**Hyper-parameters** In the experiment, we observe that knowledge distillation can improve the performance of MP and SSL, so we apply knowledge distillation to all methods including the baseline for a fair comparison. The results of removing knowledge distillation on MP and SSL are included in the Appendix. We choose the distillation weight $\lambda = 0.5$. EAP proposes an alternative way to solve the overfitting issue, so we directly use their results. For all the DNNs, we turn off the dropout layers since we find the knowledge distillation regularization will perform better. In all the experiments, we choose $\Delta q = 0.1|M|$ where $|M|$ is the number of all mask elements. For optimizing $W$, we use a pre-trained dense initialization and update $W$ by SGD with the learning rate $\eta_1 = 0.001$ and weight decay $10^{-4}$. For optimizing input mask parameters $M$, we use the Adam optimizer (Kingma & Ba, 2014) with $\eta_2 = 0.0001$ and weight decay $10^{-5}$(MNIST)/$10^{-6}$(MS-Celeb-1M). To stabilize the training process, we exponentially decay the energy budget to the target budget, and also use this trick in MP training (i.e. decaying the sparsity budget) for fair comparisons.

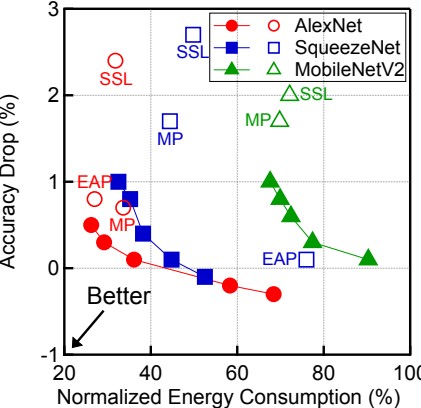

Figure 2: Accuracy drops under different energy consumptions on ImageNet.

### 6.1 IMAGENET

We set an energy budget to be less than the minimal energy consumption among the three baseline methods. We use the same performance metric (i.e. top-5 test accuracy) and hardware parameters, i.e., $e_{\text{MAC}}, e_{\text{DRAM}}, e_{\text{cache}}, e_{\text{RF}}, s_h, s_w, k_W, k_X$, as described in the EAP paper (Yang et al., 2017). We initialize all the DNNs by a pre-trained dense model, which is also used to set up the knowledge distillation regularization. The top-5 test accuracies on the dense models are 79.1% (AlexNet), 80.5% (SqueezeNet), and 90.5% (MobileNetV2). We use batch size 128 and train all the methods with 30 epochs. For SSL and NetAdapt, we apply 20 additional epochs to achieve comparable results. We implement the projection operation $\text{P}_{\Omega(E_{\text{budget}})}$ on GPU, and it takes $< 0.2s$ to perform it in our experiments. The detailed wall-clock result is included in the Appendix.

Table 1 shows the top-5 test accuracy drop and energy consumption of various methods compared to the dense model. Our training framework consistently achieves a higher accuracy with a lower

Table 1: Energy consumption and accuracy drops compared to dense models on ImageNet. We set the energy budget according to the lowest energy consumption obtained from prior art.

| DNNs | AlexNet | | | | | SqueezeNet | | | | MobileNetV2 | | |
|---|---|---|---|---|---|---|---|---|---|---|---|---|
| Energy Budget | 26% | | | | | 38% | | | | 68% | | |
| Methods | MP | SSL | EAP | NetAdapt | **Ours** | MP | SSL | EAP | **Ours** | MP | SSL | **Ours** |
| Accuracy Drop | 0.7% | 2.4% | 0.8% | 4.4% | **0.5%** | 1.7% | 2.7% | **0.1%** | 0.4% | 1.7% | 2.0% | **1.0%** |
| Energy | 34% | 32% | 27% | **26%** | 26% | 44% | 50% | 76% | **38%** | 70% | 72% | **68%** |
| Nonzero Weights Ratio | 8% | 35% | 9% | 10% | 31% | 34% | 61% | 28% | 48% | 52% | 63% | 63% |

energy consumption under the same energy budget. For instance on AlexNet, under a smaller energy budget ($26\% < 27\%$), our method achieves lower accuracy drop over EAP (0.5% vs. 0.8%). The advantage is also evident in SqueezeNet and MobileNetV2 that are already light-weight by design. EAP does not report data on MobileNetV2. We observe that weight sparsity is *not* a good proxy for energy consumption. Our method achieves lower energy consumption despite having higher density.

Figure 2 comprehensively compares our method with prior work. Solid markers represent DNNs trained from our framework under different energy budgets ($x$-axis). Empty markers represent DNNs produced from previous techniques. DNNs trained by our method have lower energies with higher accuracies (i.e., solid markers are closer to the bottom-left corner than empty markers). For instance on SqueezeNet, our most energy-consuming DNN still reduces energy by 23% while improves accuracy by 0.2% compared to EAP.

## 6.2 MNIST AND MS-CELEB-1M

MNIST and MS-Celeb-1M (Guo et al., 2016) represent datasets where inputs have regular patterns that are amenable to input masking. For instance, MS-Celeb-1M is a face image dataset and we use its aligned face images where most of the facial features are located in the center of an image. In such scenarios, training input masks lets us control the sparsity of the layer inputs and thus further reduce energy than merely pruning model parameters as in conventional methods. We do not claim that applying input mask is a general technique; rather, we demonstrate its effectiveness when applicable.

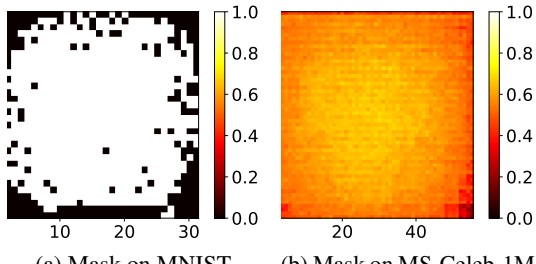

(a) Mask on MNIST    (b) Mask on MS-Celeb-1M

Figure 3: Input mask learned on MNIST and MS-Celeb-1M. For MNIST, the input mask for the first layer (with one channel) is shown. For MS-Celeb-1M, we show the input mask averaged across the 96 channels in the $7^{th}$ layer. 0 indicates a pixel is masked off, and 1 indicates otherwise.

We compare our method with MP and SSL using LeNet-5 and MobileNetV2 for these two datasets, respectively. The pre-trained dense LeNet-5 has 99.3% top-1 test accuracy on MNIST, and the dense MobileNetV2 has 65.6% top-5 test accuracy on MS-Celeb-1M. EAP does not report data on these two datasets. Similar to the evaluation on ImageNet, we set the energy budget to be lower than the energy consumptions of MP and SSL. We use batch size 32 on MNIST and 128 on MS-Celeb-1M, and number of epochs is set the same as the ImageNet experiments. Table 2 compares the energy consumption and accuracy drop. Our method consistently achieves higher accuracy with lower energy under the same or even smaller energy budget. We visualize the sparsity of the learned input masks in Figure 3.

Table 2: Energy consumptions and accuracy drops on MNIST and MS-Celeb-1M.

| DNNs@Dataset | LeNet-5@MNIST | | | | | | MobileNetV2@MS-Celeb-1M | | |
|---|---|---|---|---|---|---|---|---|---|
| Energy Budget | 17% | | | | | | 60% | | |
| Methods | MP | SSL | NetAdapt | SBP | BC | **Ours** | MP | SSL | **Ours** |
| Accuracy Drop | 1.5% | 1.5% | 0.6% | 1.5% | 2.2% | **0.5%** | 1.1% | 0.7% | **0.2%** |
| Energy | 18% | 20% | 18% | 22% | 26% | **17%** | 66% | 72% | **60%** |

## 7 CONCLUSION

This paper demonstrates that it is possible to train DNNs with quantitative energy guarantees in an end-to-end fashion. The enabler is an energy model that relates the DNN inference energy to the DNN parameters. Leveraging the energy model, we augment the conventional DNN training with an energy-constrained optimization process, which minimizes the accuracy loss under the constraint of a given energy budget. Using an efficient algorithm, our training framework generates DNNs with higher accuracies under the same or lower energy budgets compared to prior art.

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

APPENDICES

DETAIL OF ENERGY CONSUMPTION FOR DATA ACCESS

FULLY CONNECTED LAYER

To multiply $X^{(v)} \in \mathbb{R}^c$ and $W^{(v)} \in \mathbb{R}^{c \times d}$, each nonzero element of $W^{(v)}$ is used once but loaded three times, once each from DRAM, cache and RF, respectively. Thus, the number of DRAM, cache, and RF accesses for weight matrix $W^{(v)}$ is:

$$N_{\text{DRAM}}^{\text{weights}} = N_{\text{cache}}^{\text{weights}} = N_{\text{RF}}^{\text{weights}} = \|W^{(v)}\|_0. \tag{15}$$

Input $X^{(v)}$ is fed into the systolic array $\lceil d/s_w \rceil$ times, where $s_w$ denotes the the systolic array width. Thus, the number of cache accesses for $X^{(v)}$ is:

$$N_{\text{cache}}^{\text{input}} = \lceil d/s_w \rceil \|X^{(v)}\|_0. \tag{16}$$

Let $k_X$ be the cache size for input $X^{(v)}$. If $k_X$ is less than $\|X^{(v)}\|_0$, there are $\|X^{(v)}\|_0 - k_X$ elements that must be reloaded from DRAM every time. The rest $k_X$ elements need to load from only DRAM once as they will always reside in low-level memories. Thus, there are $\lceil d/s_w \rceil (\|X^{(v)}\|_0 - k_X) + k_X$ DRAM accesses for $X^{(v)}$. In addition, the output vector of the FC layer (result of $X^{(v)}W^{(v)}$) needs to be written back to DRAM, which further incurs $d$ DRAM accesses. Thus, The total number of DRAM accesses to retrieve $X^{(v)}$ is:

$$N_{\text{DRAM}}^{\text{input}} = \lceil d/s_w \rceil \max(0, \|X^{(v)}\|_0 - k_X) + \min(k_X, \|X^{(v)}\|_0) + d. \tag{17}$$

Each input element is loaded from RF once for each MAC operation, and there are two RF accesses incurred by accumulation for each MAC operation (one read and one write). Thus, the total number of RF accesses related to $X^{(v)}$ is:

$$N_{\text{RF}}^{\text{input}} = d\|X^{(v)}\|_0 + 2\|W^{(v)}\|_0. \tag{18}$$

In summary, the data access energy of a fully connected layer $v$ is expressed as follows, in which each component follows the derivations in Equation (15) through Equation (18):

$$E_{\text{data}}^{(v)} = e_{\text{DRAM}}(N_{\text{DRAM}}^{\text{input}} + N_{\text{DRAM}}^{\text{weights}}) + e_{\text{cache}}(N_{\text{cache}}^{\text{input}} + N_{\text{cache}}^{\text{weights}}) + e_{\text{RF}}(N_{\text{RF}}^{\text{input}} + N_{\text{RF}}^{\text{weights}}). \tag{19}$$

CONVOLUTION LAYER

Similar to a FC layer, the data access energy of a CONV layer $u$ is modeled as:

$$E_{\text{data}}^{(u)} = e_{\text{DRAM}}(N_{\text{DRAM}}^{\text{input}} + N_{\text{DRAM}}^{\text{weights}}) + e_{\text{cache}}(N_{\text{cache}}^{\text{input}} + N_{\text{cache}}^{\text{weights}}) + e_{\text{RF}}(N_{\text{RF}}^{\text{input}} + N_{\text{RF}}^{\text{weights}}). \tag{20}$$

The notations are the same as in FC layer. We now show how the different components are modeled.

To convolve $W^{(u)} \in \mathbb{R}^{d \times c \times r \times r}$ with $X^{(u)} \in \mathbb{R}^{c \times h \times w}$, each nonzero element in the weight tensor $W^{(u)}$ is fed into the systolic array $\lceil h'w'/s_h \rceil$ times, where $s_h$ denotes the height of the systolic array and $h'$ and $w'$ are dimension parameters of $X^{(u)}$. Thus,

$$N_{\text{cache}}^{\text{weights}} = \lceil h'w'/s_h \rceil \|W^{(u)}\|_0. \tag{21}$$

Similar to the FC layer, the number of RF accesses for $W^{(u)}$ during all the MAC operations is:

$$N_{\text{RF}}^{\text{weights}} = h'w'\|W^{(u)}\|_0. \tag{22}$$

Let $k_W$ be the cache size for the weight matrix $W^{(u)}$. If $\|W^{(u)}\|_0 > k_W$, there are $k_W$ nonzero elements of $W^{(u)}$ that would be accessed from DRAM only once as they would reside in the cache, and the rest $\|W^{(u)}\|_0 - k_W$ elements would be accessed from DRAM by $\lceil h'w'/s_h \rceil$ times. Thus,

$$N_{\text{DRAM}}^{\text{weights}} = \lceil h'w'/s_h \rceil \max(0, \|W^{(u)}\|_0 - k_W) + \min(k_W, \|W^{(u)}\|_0). \tag{23}$$

Let $k_X$ be the cache size for input $X^{(u)}$. If every nonzero element in $X^{(u)}$ is loaded from DRAM to cache only once, $N_{\text{DRAM}}^{\text{input}}$ would simply be $\|X^{(u)}\|_0$. In practice, however, the cache size $k_X$ is much

smaller than $\|X^{(u)}\|_0$. Therefore, some portion of $X^{(u)}$ would need to be re-loaded. To calculate the amount of re-loaded DRAM access, we observe that in real hardware $X^{(u)}$ is loaded from DRAM to the cache at a row-granularity.

When the input $X^{(u)}$ is dense, there are at least $cw$ elements loaded at once. In this way, the cache would first load $\lfloor k_X/(cw)\rfloor$ rows from DRAM, and after the convolutions related to these rows have finished, the cache would load the next $\lfloor k_X/(cw)\rfloor$ rows in $X^{(u)}$ for further processing. The rows loaded in the above two rounds have overlaps due to the natural of the convolution operation. The number of overlaps $R_{\text{overlap}}$ is $\lceil h/(\lfloor k_X/cw\rfloor - r + s)\rceil - 1$, and each overlap has $cw(r-s)$ elements. Thus, $R_{\text{overlap}} \times cw(r-s)$ elements would need to be reloaded from DRAM. Finally, storing the outputs of the convolution incurs an additional $dh'w'$ DRAM writes. Summing the different parts together, the upper bound ($X^{(u)}$ is dense) number of DRAM accesses for $X^{(u)}$ is:

$$N_{\text{DRAM}}^{\text{input}} = \|X^{(u)}\|_0 + (\lceil h/(\lfloor k_X/cw\rfloor - r + s)\rceil - 1)cw(r-s) + dh'w'. \tag{24}$$

When the input $X^{(u)}$ is not dense, we can still count the exact number of elements in the overlaps $N_{\text{overlap}}$ of the consecutive loading rounds, so we have:

$$N_{\text{DRAM}}^{\text{input}} = \|X^{(u)}\|_0 + N_{\text{overlap}} + dh'w'. \tag{25}$$

Every nonzero element in the unfolded input $\bar{X}^{(u)}$ would be fed into the systolic array $\lceil d/s_w\rceil$ times (for grouped convolution, this number is divided by the number of groups). Each MAC operation introduces 2 RF accesses. Thus,

$$N_{\text{cache}}^{\text{input}} = \lceil d/s_w\rceil\|\bar{X}\|_0, \; N_{\text{RF}}^{\text{input}} = d\|\bar{X}^{(u)}\|_0 + 2h'w'\|W^{(u)}\|_0. \tag{26}$$

PROOF TO THEOREM 1

*Proof.* First, it is easy to see that (10) is equivalent to the following problem

$$\max_{\xi \text{ is binary}} \langle Z \odot Z, \xi\rangle, \quad \text{s.t.} \quad \xi \in \Omega(E_{\text{budget}}). \tag{27}$$

Note that if the optimal solution to problem (27) is $\bar{\xi}$, the solution to problem (10) can be obtained by $Z \odot \bar{\xi}$; given the solution to (10), the solution to (27) can be obtained similarly.

Therefore, we only need to prove that (27) is equivalent to (12). Meeting the following two conditions guarantees that (27) and (12) are equivalent since they have identical objective functions:

1. Any optimal solution of problem (27) is in the constraint set of problem (12);

2. Any optimal solution of problem (12) is in the constraint set of problem (27).

Let us prove the first condition. Let $\hat{\xi}$ be the optimal solution to (27). Then for any $u \in U \cup V$, the elements of $Z^{(u)}$ selected by $\hat{\xi}^{(u)}$ are the largest (in terms of magnitude) $\|\hat{\xi}^{(u)}\|_0$ elements of $Z^{(u)}$; otherwise there would exist at least one element that can be replaced by another element with a larger magnitude, which would increase the objective value in (27). Since $\hat{\xi} \in \Omega(E_{\text{budget}})$, according to the definition of $A$ in (13), $\hat{\xi}$ satisfies the constraint of (12).

Let us now prove the second condition. The definition of $A$ in (13) show that there could at most be two different $A^{(u)}$ values for each element $u$, and the largest $k$ elements in $Z^{(u)}$ always have the smaller value, i.e., $\alpha_1^{(u)} + \alpha_3^{(u)}$. Let $\bar{\xi}$ be the optimal solution to the knapsack problem (12). For any $u \in U \cup V$, the elements selected by $\bar{\xi}^{(u)}$ are also the largest elements in $Z^{(u)}$ in terms of magnitude; otherwise there would exist an element $Z_j^{(u)}$ that has a larger magnitude but corresponds to a smaller $A_j^{(u)}$ ((13) shows that $A_i^{(u)} \geq A_j^{(u)}$ when $|Z_i^{(u)}| \leq |Z_j^{(u)}|$). This would contradict the fact that $\bar{\xi}$ is optimal. In addition, $\bar{\xi}$ meets the constraint in problem (12). Therefore, $\bar{\xi} \in \Omega(E_{\text{budget}})$.

It completes the proof. $\qquad\square$

AN $(1 + \epsilon)$-APPROXIMATE SOLUTION FOR PROBLEM (10)

**Theorem 3.** *For the projection problem* (10)*, there exists an efficient approximation algorithm that has a computational complexity of $O\left(\left(n + \frac{(|U|+|V|)^3}{\epsilon^2}\right)\log\frac{n\max(A)}{\min(A_+)}\right)$ and generates a solution $W' \in \Omega(E_{\text{budget}})$ that admits*

$$\|W' - Z\|^2 \le \left\|P_{\Omega\left(\frac{E_{\text{budget}}}{1+O(\epsilon)}\right)}(Z) - Z\right\|^2, \tag{28}$$

*where $\min(A_+)$ is the minimum of the positive elements in A.*

$|U|$ and $|V|$ denote the number of CONV and FC layers, respectively. They are very small numbers that can be treated as constants here. Thus, the computational complexity for our problem is reduced to $\tilde{O}(n + \frac{1}{\epsilon^2})$, where $\tilde{O}$ omits the logarithm term. In the following, we will prove this theorem by construction.

PROBLEM FORMULATION

**Definition 1. Inverted knapsack problem.** *Given $n$ objects $I := \{(v_i, w_i)\}_{i=1}^n$ each with weight $w_i > 0$, and value $v_i \ge 0$, define $h_I(x)$ to be the smallest weight budget to have the total value $x$:*

$$h_I(x) := \min_{\xi \in \{0,1\}^n} \sum_{i=1}^n w_i \xi_i \tag{29}$$

$$\text{s. t. } \sum_{i=1}^n v_i \xi_i \ge x$$

We are more interested in the case that the weights of $n$ objects are in $m$ clusters, i.e. there are only $m$ distinct weights,

$$|\{w_i\}_{i=1}^n| = m.$$

In our case, $m$ is proportional to the number of layers in DNN, and $n$ is the number of all the learnable weights in $W$, so $m \ll n$.

**Definition 2. Inverse of step function.** *The inverse of the step function $f$ is defined as the maximal $x$ having the function value $y$:*

$$f^{-1}(y) := \max_{f(x) \le y} x \tag{30}$$

**Observation** The inverse of the step function $h_I^{-1}(y)$ is just the maximal value we can get given the weight budget, i.e. the original knapsack problem:

$$h_I^{-1}(y) = \max_{\xi \in \{0,1\}^n} \sum_{i=1}^n v_i \xi_i, \quad \text{s. t. } \sum_{i=1}^n w_i \xi_i \le y. \tag{31}$$

**Observation** Given a step function with $l$ breakpoints, its inverse can be generated with $O(l)$ time complexity, and vice versa.

Thus, given the step function of $h_I$ in (29) which has $l$ breakpoints, we can get $h_I^{-1}$ (i.e. the original knapsack problem) within $O(l)$ time complexity.

**Definition 3. $w$-uniform.** *Step function $f$ is $w$-uniform if the ranges of $f$ is from $-\infty, 0, w, 2w, ..., lw$.*

**Observation** If all the objects in $I$ have the same weight $w$, i.e. $m = 1$, then the function $h_I(x)$ is nondecreasing and $w$-uniform. Moreover, its breakpoints are:

$$(0,0), (v_1, w), (v_1 + v_2, 2w), ..., \left(\sum_{i=1}^n v_i, nw\right),$$

if the objects' indices follows the decreasing order in terms of the values, i.e. $v_1 \geq v_2 \geq ... \geq v_n$. Thus we can get all possible function values of $h_I(x)$:

$$h_I(x) = kw, \quad \forall x \in \left( \sum_{i=1}^{k-1} v_i, \sum_{i=1}^{k} v_i \right].$$

**Definition 4. (min, +)-convolution.** *For functions $f, g$, the (min, +)-convolution is:*

$$(f \oplus g)(x) = \min_{x'}(f(x') + g(x - x')).$$

**Observation** If object sets $I_1 \cap I_2 = \emptyset$, then

$$f_{I_1 \cup I_2} = f_{I_1} \oplus f_{I_2}.$$

**Observation** The inverse of (min, +)-convolution between $w$-uniform function $f$ and $w$-uniform function $g$ is the (max, +)-convolution between $f^{-1}$ and $g^{-1}$:

$$(f \oplus g)^{-1}(y) = \max_{y' \in \{0, 1w, ..., lw\}}(f^{-1}(y') + g^{-1}(y - y')). \tag{32}$$

**Lemma 4.** *For any $f$ and $g$ nonnegative step functions, given an arbitrary number $b$, we always have*

$$\min\{f \oplus g, b\} = \min\{\min\{f, b\} \oplus \min\{g, b\}, b\} \tag{33}$$

*Proof.* Given any $x$, let $z \in \operatorname{Arg\,min}_{x'} f(x') + g(x - x')$ and $\bar{z} \in \operatorname{Arg\,min}_{x'} \min(f(x'), b) + \min(g(x - x'), b)$, so we have $(f \oplus g)(x) = f(z) + g(x - z)$ and $(\min\{f, b\} \oplus \min\{g, b\})(x) = \min(f(\bar{z}), b) + \min(g(x - \bar{z}), b)$.

Consider the following cases:

1. $(f \oplus g)(x) \geq b$. In this case, we claim that $(\min\{f, b\} \oplus \min\{g, b\})(x) \geq b$. We prove it by contradiction. Suppose $(\min\{f, b\} \oplus \min\{g, b\})(x) < b$ which implies $\min(f(\bar{z}), b) + \min(g(x - \bar{z}), b) < b$. Because both $f$ and $g$ are nonnegative, we have $f(\bar{z}) < b$ and $g(x - \bar{z}) < b$ which imply $\min(f(\bar{z}), b) + \min(g(x - \bar{z}), b) = f(\bar{z}) + g(x - \bar{z}) < b$, However, this contradicts $(f \oplus g)(x) \geq b$. Therefore, we have $\min((f \oplus g)(x), b) = \min((\min\{f, b\} \oplus \min\{g, b\})(x), b) = b$.

2. $(f \oplus g)(x) < b$. In this case, we have $f(z) < b$ and $g(x - z) < b$, so $\min(f(\bar{z}), b) + \min(g(x - \bar{z}), b) \leq \min(f(z), b) + \min(g(x - z), b) = f(z) + g(x - z) = (f \oplus g)(x) < b$. Since both $f$ and $g$ are nonnegative, we have $f(\bar{z}) < b$ and $g(x - \bar{z}) < b$ which imply $\min(f(\bar{z}), b) + \min(g(x - \bar{z}), b) = f(\bar{z}) + g(x - \bar{z}) \geq (f \oplus g)(x)$. Therefore, we have $\min(f(\bar{z}), b) + \min(g(x - \bar{z}), b) = f(z) + g(x - z) \Leftrightarrow (\min\{f, b\} \oplus \min\{g, b\})(x) = (f \oplus g)(x)$.

$\square$

EFFICIENCY OF (MIN, +)-CONVOLUTION

**Lemma 5.** *Let $f$ and $g$ be nondecreasing $w$-uniform functions with $O(l)$ breakpoints, the (min, +)-convolution $f \oplus g$ (having $O(l)$ breakpoints) can be generated with $O(l^2)$ time complexity.*

*Proof.* Firstly, we compute the inverse representation of $f$ and $g$, i.e. compute $f^{-1}$ and $g^{-1}$ from Equation (30). The inverse representation can be computed in $O(l)$ time (proportional to the number of breakpoints). From Equation (32), we can compute the inverse of $f \oplus g$. For each $y \in \{0, 1w, ..., 2lw\}$, function $(f \oplus g)^{-1}(y)$ can be computed in $O(l)$ time by brute force. Thus a total $O(l^2)$ is enough to get $(f \oplus g)^{-1}$ which has $O(l)$ breakpoints. We can get $f \oplus g$ via $(f \oplus g)^{-1}$ by the inverse definition (30) in $O(l)$ time. $\square$

**Lemma 6.** *Let $f$ and $g$ be nondecreasing step functions with $l$ breakpoints in total, $\min\{f \oplus g, b\}$ can be approximated by a step function $\phi_b$ with $O(l + \frac{1}{\epsilon^2})$ complexity and $2\epsilon b$ additive error, i.e. $\min\{f \oplus g, b\} \leq \phi_b \leq \min\{f \oplus g, b\} + 2\epsilon b$. The resultant function $\phi_b$ has $O(1/\epsilon)$ breakpoints.*

*Proof.* We can construct $(\epsilon b)$-uniform functions $f'_b, g'_b$ which have $\lceil 1/\epsilon \rceil$ breakpoints:

$$f'_b(x) = \left\lceil \frac{\min(b, f(x))}{\epsilon b} \right\rceil \epsilon b, \quad g'_b(x) = \left\lceil \frac{\min(b, g(x))}{\epsilon b} \right\rceil \epsilon b.$$

This needs $O(l)$ computational complexity. From Lemma 5, we can compute $f'_b \oplus g'_b$ with $O(\frac{1}{\epsilon^2})$ time complexity and $\phi_b = \min\{f'_b \oplus g'_b, b\}$ has $O(1/\epsilon)$ breakpoints. Because $f'_b$ and $g'_b$ are constructed by ceiling $\min\{f, b\}$ and $\min\{g, b\}$, we have:

$$\min\{f, b\} \oplus \min\{g, b\} \leq f'_b \oplus g'_b \leq \min\{f, b\} \oplus \min\{g, b\} + 2\epsilon b,$$

which implies

$$\min\{\min\{f, b\} \oplus \min\{g, b\}, b\} \leq \min\{f'_b \oplus g'_b, b\} \leq \min\{\min\{f, b\} \oplus \min\{g, b\}, b\} + 2\epsilon b.$$

From Lemma 4, we know that $\min\{\min\{f, b\} \oplus \min\{g, b\}, b\} = \min\{f \oplus g, b\}$, so it completes the proof. □

**Lemma 7.** *Let $f_1, f_2, ..., f_m$ be nondecreasing step functions with $l$ breakpoints in total, $\min\{f_1 \oplus f_2 \oplus ... \oplus f_m, b\}$ can be approximated by a step function $\psi_b$ with $O(l + m/\epsilon^2)$ computational complexity and $m\epsilon b$ additive error. The resultant function $\psi_b$ has $O(1/\epsilon)$ breakpoints.*

*Proof.* From Lemma 6, we have shown the case $m = 2$. For general $m > 2$, we can construct a binary tree to approximate pairs of functions, e.g., if $m = 4$, we can firstly approximate $\psi^{(1)} \approx \min\{f_1 \oplus f_2, b\}$, and $\psi^{(2)} \approx \min\{f_3 \oplus f_4, b\}$, then approximate $\psi_b^{(3)} \approx \min\{\psi^{(1)} \oplus \psi^{(2)}, b\}$.

By this way, we construct a binary tree which has $O(\log m)$ depth and $O(m)$ nodes. In the beginning, we use ceil function to construct $m$ new $\epsilon b$-uniform functions:

$$f'_{i,b}(x) = \left\lceil \frac{\min(b, f_i(x))}{\epsilon b} \right\rceil \epsilon b, \forall i \in \{1, 2, ..., m\}.$$

Then we can use the binary tree to "merge" all the $m$ functions in pairs, via $O(\log m)$ iterations. Without loss of generality, we assume $m$ is a power of two. We can recursively merge $t$ functions into $t/2$ functions:

1. Initialize $t = m$, $g'_{i,b} = f'_{i,b}, \forall i \in \{1, ..., t\}$.

2. Reassign $g'_{i,b} = \min\{g'_{2i-1,b} \oplus g'_{2i,b}, b\}, \forall i \in \{1, ..., t/2\}$. According to Lemma 6, the number of break points of $\min\{g'_{2i-1,b} \oplus g'_{2i,b}, b\}$ is still $O(1/\epsilon)$.

3. $t = t/2$. If $t > 1$, go back to Step 2.

4. Return $\psi_b := \min\{g'_{1,b}, b\}$.

For this binary tree, functions of the bottom leaf nodes have $\epsilon b$ additive error, and every (min, +)-convolution $f' \oplus g'$ will accumulate the additive error from the two functions $f'$ and $g'$. The root node of the binary tree will accumulate the additive errors from all the $m$ leaf nodes, thus the resultant function $\psi_b \leq \min\{f_1 \oplus ... \oplus f_m, b\} + m\epsilon b$. For the computational complexity, initializing $f'_{i,b}$ takes $O(l)$, Step 1 takes $O(l)$, Step 2 and 3 take $O(m/\epsilon^2)$ (since there are $O(m)$ nodes in the binary tree), and Step 4 takes $O(m/\epsilon)$. Therefore, there is $O(l + m/\epsilon^2)$ in total. □

**Lemma 8.** *For the inverted knapsack problem defined in Equation (29), if all the $n$ objects can be separated into $m$ groups $I_1, ..., I_m$ which have $m$ distinct weights, there exists an approximate algorithm with computational complexity $O((n + \frac{m^3}{\epsilon^2}) \log \frac{n \max(w)}{\min(w)})$ which can approximate $h_I$ by $\tilde{h}_I$:*

$$h_I(x) \leq \tilde{h}_I(x) \leq (1 + O(\epsilon))h_I(x), \quad \forall x.$$

*Proof.* Firstly, the step function $h_{I_i}, \forall i \in \{1, 2, ..., m\}$ can be easily generated within $O(n \log n)$ by sorting the objects of each group according to their values (in descending order). From the definition of (min, +)-convolution, we know that $h_I = h_{I_1} \oplus ... \oplus h_{I_m}$. Let us construct an algorithm to approximate $h_I$:

1. Construct a set $\mathcal{B} := \{2^i n \max(w) \in [\min(w), n \max(w)]; i \in \mathbb{Z}_{\leq 0}\}$, where $\min(w)$ and $\max(w)$ are the minimum and maximum weight of items respectively, and $\mathbb{Z}_{\leq 0}$ is the nonpositive integer set. We have $|\mathcal{B}| = O(\log \frac{n \max(w)}{\min(w)})$.

2. For every $b \in \mathcal{B}$, construct $\psi_b$ to approximate $\min\{h_{I_1} \oplus ... \oplus h_{I_m}, b\}$ based on Lemma 7.

3. Construct function $\tilde{h}_I^{-1}$:

$$\tilde{h}_I^{-1}(y) = \begin{cases} \psi_b^{-1}(y), & \text{if } b/2 < y \leq b \text{ and } y > \min(\mathcal{B}); \\ \psi_{\min(\mathcal{B})}^{-1}(y), & \text{if } y \leq \min(\mathcal{B}). \end{cases}$$

where $\min(\mathcal{B})$ is the minimum element in $\mathcal{B}$. The resultant function $\tilde{h}_I^{-1}$ (or $\tilde{h}_I$) has at most $O(\frac{1}{\epsilon} \log \frac{n \max(w)}{\min(w)})$ breakpoints.

4. Compute the original function $\tilde{h}_I$ from $\tilde{h}_I^{-1}$.

According to the above procedure, for any $h_I(x) \in (b/2, b]$, $\tilde{h}_I(x)$ approximate $h_I(x)$ with additive error $O(m\epsilon b)$, so we have $h_I(x) \leq \tilde{h}_I(x) \leq (1 + O(m\epsilon))h_I(x)$. The algorithm takes $O((n + m/\epsilon^2) \log \frac{n \max(w)}{\min(w)})$, if we require the approximation factor to be $1 + O(\epsilon)$, i.e.,

$$h_I(x) \leq \tilde{h}_I(x) \leq (1 + O(\epsilon))h_I(x), \quad \forall x,$$

we need

$$O\left((n + m^3/\epsilon^2) \log \frac{n \max(w)}{\min(w)}\right)$$

time complexity. $\qquad\qquad\square$

**Theorem 9.** *For the knapsack problem defined in Equation (12), if all the $n$ objects have $m$ distinct weights, there exists an approximate algorithm with computational complexity $O((n + \frac{m^3}{\epsilon^2}) \log \frac{n \max(w)}{\min(w)})$ to generate a function $\tilde{h}_I^{-1}$ satisfying:*

$$h_I^{-1}\left(\frac{y}{1 + O(\epsilon)}\right) \leq \tilde{h}_I^{-1}(y) \leq h_I^{-1}(y), \quad \forall y.$$

*Proof.* From Lemma 8, we have $\tilde{h}_I(x) \leq (1 + O(\epsilon))h_I(x)$ which implies

$$\{x \mid (1 + O(\epsilon))h_I(x) \leq y\} \subseteq \{x \mid \tilde{h}_I(x) \leq y\}.$$

So

$$\max_{h_I(x) \leq y/(1+O(\epsilon))} x \leq \max_{\tilde{h}_I(x) \leq y} x \Leftrightarrow h_I^{-1}\left(\frac{y}{1 + O(\epsilon)}\right) \leq \tilde{h}_I^{-1}(y).$$

Similarly, we can get $\{x \mid \tilde{h}_I(x) \leq y\} \subseteq \{x \mid h_I(x) \leq y\}$ from Lemma 8, so we have

$$\max_{h_I(x) \leq y} x \geq \max_{\tilde{h}_I(x) \leq y} x \Leftrightarrow h_I^{-1}(y) \geq \tilde{h}_I^{-1}(y).$$

$\qquad\qquad\square$

Let $I$ be the set of objects whose weights are nonzero elements in $A$ and values are the corresponding elements in $Z \odot Z$, i.e. $I_+ = \{(Z_i^2, A_i) \mid \forall i \in \{1, 2, ..., |A|\} \text{ and } A_i > 0\}$, $\tilde{\xi}^+$ be the solution corresponding to $\tilde{h}_I^{-1}(E_{\text{budget}} - \sum_{u \in U \cup V} \alpha_4^{(u)})$. Let $\tilde{\xi}_{I_+^c} = \mathbf{1}$ and $\tilde{\xi}_{I_+} = \tilde{\xi}^+$, where $I_+^c = \{(Z_i^2, A_i) \mid \forall i \in \{1, 2, ..., |A|\} \text{ and } A_i = 0\}$ is the complement of $I_+$. Here we have $m \leq 2|U| + |V|$ distinct values in $A$. According to Theorem 9, we have $\langle Z \odot Z, \tilde{\xi}\rangle \geq \max_\xi \langle Z \odot Z, \xi\rangle$, s.t. $\langle A, \xi\rangle \leq \frac{E_{\text{budget}} - \sum_{u \in U \cup V} \alpha_4^{(u)}}{1 + O(\epsilon)}$, which implies

$$\langle Z \odot Z, \tilde{\xi}\rangle \geq \max_{\xi \in \Omega(E_{\text{budget}}/(1+O(\epsilon)))} \langle Z \odot Z, \xi\rangle.$$

From Theorem 9, we can directly get Theorem 3.

---

**Algorithm 2:** Greedy Algorithm to Solve Problem (12).

---

**Input:** $Z, A, E_{\text{budget}}, \{\alpha^{(u)}\}_{u \in U \cup V}$ as in (12).

**Result:** Greedy solution $\tilde{\xi}$ for problem (12).

1 Initialize $b = 0, \xi = \mathbf{0}$.

2 Generate the profit density $\delta$:

$$\delta_j = \begin{cases} (Z_j)^2/A_j, \text{ if } A_j > 0; \\ \infty, \text{ if } A_j = 0. \end{cases}$$

3 Sort $\delta$, let $I$ be the indices list of the sorted $\delta$ (in descending order).

4 **foreach** *index* $j \in I$ **do**

5      $b = b + A_j$;

6      If $b > E_{\text{budget}} - \sum_{u \in U \cup V} \alpha_4^{(u)}$, exit loop;

7      $\xi_j = 1$;

8 **end**

9 $\tilde{\xi} = \xi$.

---

PROOF TO THEOREM 2

*Proof.* From Theorem 1, we know the original projection problem (10) is equivalent to the knapsack problem (12). So proving the inequality (14) is equivalent to proving

$$\langle Z \odot Z, \tilde{\xi} \rangle \geq \langle Z \odot Z, \xi^* \rangle - \text{Top}_{\|\tilde{\xi}\|_0 + 1}((Z \odot Z) \oslash A) \cdot R(\tilde{\xi}) \tag{34}$$

and

$$\langle Z \odot Z, \tilde{\xi} \rangle \geq \langle Z \odot Z, \xi^* \rangle - \text{Top}_{\|\tilde{\xi}\|_0 + 1}((Z \odot Z) \oslash A) \cdot (\max(A) - \gcd(A)), \tag{35}$$

where $\tilde{\xi}$ is the greedy solution of knapsack problem corresponding to $W''$, and $\xi^*$ is the exact solution of knapsack problem corresponding to $\text{P}_{\Omega(E_{\text{budget}})}(Z)$, i.e.,

$$W'' = Z \odot \tilde{\xi}, \quad \text{P}_{\Omega(E_{\text{budget}})}(Z) = Z \odot \xi^*.$$

Firstly, let us prove the inequality (35). If we relax the values of $\xi$ to be in the range $[0, 1]$ instead of $\{0, 1\}$, the discrete constraint is removed so that the constraint set becomes

$$\Delta = \left\{ \xi \mid \mathbf{0} \leq \xi \leq \mathbf{1} \text{ and } \langle A, \xi \rangle \leq E_{\text{budget}} - \sum_{u \in U \cup V} \alpha_4^{(u)} \right\}.$$

So the $0/1$ knapsack problem is relaxed as a linear programming. This relaxed problem is called fractional knapsack problem, and there is a greedy algorithm (Dantzig, 1957) which can exactly solve the fractional knapsack problem. Slightly different from our Algorithm 2, the greedy algorithm for the fractional knapsack can select a fraction of the item, so its remaining budget is always zero. The optimal objective value of the fractional knapsack is

$$\max_{\xi \in \Delta} \langle Z \odot Z, \xi \rangle = \langle Z \odot Z, \tilde{\xi} \rangle + \text{Top}_{\|\tilde{\xi}\|_0 + 1}((Z \odot Z) \oslash A) \cdot R(\tilde{\xi}).$$

Since the constraint set of the fractional knapsack problem is a superset of the constraint of the original knapsack problem, we have $\langle Z \odot Z, \xi^* \rangle \leq \max_{\mathbf{0} \leq \xi \leq \mathbf{1}} \langle Z \odot Z, \xi \rangle$, that leads to inequality (34).

Secondly, we show that the inequality (35) is also true. Since all the coefficients in $A$ are multiples of $\gcd(A)$, we can relax the original $0/1$ knapsack problem in this way: for each item, split them to several items whose coefficients in the constraint are $\gcd(A)$, and the coefficients in the objective function are split equally. For the $j$-th item, the coefficient in the constraint is $A_j$ and the coefficient in the objective function is $(Z \odot Z)_j$. It will be split into $A_j/\gcd(A)$ items, and the $j$-th item is associated with coefficient $(Z_j^2/A_j) \cdot \gcd(A)$ in the objective function. This relaxation gives us a new $0/1$ knapsack problem, where all the items have the same coefficient in the constraint, so the optimal solution is just selecting the ones with the largest coefficients in the objective function. We can formulate this problem as a relaxed knapsack problem by replacing the constraint of $\xi$ into $\xi \in \Gamma$, where

$$\Gamma = \left\{ \xi \mid \text{for all } j, \xi_j \text{ is a multiple of } \frac{\gcd(A)}{A_j}, 0 \leq \xi_j \leq 1, \text{ and } \langle A, \xi \rangle \leq E_{\text{budget}} - \sum_{u \in U \cup V} \alpha_4^{(u)} \right\}.$$

All the elements of the solution are either $0$ or $1$ except the last picked one which corresponds to $\text{Top}_{\|\tilde{\xi}\|_0+1}((Z \odot Z) \oslash A)$. Let the $(\|\tilde{\xi}\|_0 + 1)$-th largest element in $(Z \odot Z) \oslash A$ be indexed by $t$. We have $0 \leq \tilde{\xi}_t \leq 1 - \gcd(A)/A_t$. Therefore, comparing with the original $0/1$ knapsack problem, we have

$$
\begin{aligned}
\max_{\xi \in \Gamma} \langle Z \odot Z, \xi \rangle &\leq \langle Z \odot Z, \tilde{\xi} \rangle + (Z \odot Z)_t \cdot (1 - \gcd(A)/A_t) \\
&= \langle Z \odot Z, \tilde{\xi} \rangle + \text{Top}_{\|\tilde{\xi}\|_0+1}((Z \odot Z) \oslash A) \cdot A_t \cdot (1 - \gcd(A)/A_t) \\
&= \langle Z \odot Z, \tilde{\xi} \rangle + \text{Top}_{\|\tilde{\xi}\|_0+1}((Z \odot Z) \oslash A) \cdot (A_t - \gcd(A)) \\
&\leq \langle Z \odot Z, \tilde{\xi} \rangle + \text{Top}_{\|\tilde{\xi}\|_0+1}((Z \odot Z) \oslash A) \cdot (\max(A) - \gcd(A))
\end{aligned}
$$

Since $\{\xi \mid \xi \text{ is binary}\} \subseteq \Gamma$, we have $\langle Z \odot Z, \xi^* \rangle \leq \max_{\xi \in \Gamma} \langle Z \odot Z, \xi \rangle$. So we have the inequality (35).  □

## SUPPLEMENTARY EXPERIMENT RESULTS

### RESULTS OF BASELINE WITHOUT KNOWLEDGE DISTILLATION

Table 3 shows the energy and accuracy drop results of the baseline methods MP and SSL when the knowledge distillation is removed from their loss function. By using knowledge distillation, the results in Table 1 are much better. Therefore, we use knowledge distillation in all the experiments when it is applicable.

Table 3: Energy consumption and accuracy drops compared to dense models on ImageNet. Knowledge distillation is removed from the loss function.

| DNNs | AlexNet | | SqueezeNet | | MobileNetV2 | |
|---|---|---|---|---|---|---|
| Methods | MP | SSL | MP | SSL | MP | SSL |
| Accuracy Drop | 2.6% | 17.6% | 1.9% | 16.0% | 2.0% | 1.9% |
| Energy | 34% | 35% | 44% | 52% | 71% | 77% |

### ENERGY-CONSTRAINED PROJECTION EFFICIENCY

The projection operation $P_{\Omega(E_{\text{budget}})}$ in Algorithm 1 can be implemented on GPU. We measured its wall-clock time on a GPU server (CPU: Xeon E3 1231-v3, GPU: GTX 1080 Ti), and the result is shown in Table 4 (the time is averaged over 100 iterations).

Table 4: Wall-clock time of the projection operation $P_{\Omega(E_{\text{budget}})}$.

| DNNs | AlexNet | SqueezeNet | MobileNetV2 |
|---|---|---|---|
| Time (seconds) | 0.170 | 0.023 | 0.032 |

