# OpenReview forum: "Energy-Constrained Compression for Deep Neural Networks via Weighted Sparse Projection and Layer Input Masking"
_ICLR.cc/2019/Conference_

### Official Review · AnonReviewer3 · 2018-11-01
**Interesting paper**

**Rating:** 7
**Confidence:** 4

**Review:**

The paper proposes a method for neural network training under a hard energy constraint (i.e. the method guarantees the energy consumption to be upper bounded). Based on a systolic array hardware architecture the authors model the energy consumption of transferring the weights and activations into different levels of memory (DRAM, Cache, register file) during inference. The energy consumption is therefore determined by the number of nonzero elements in the weight and activation tensors. To minimize the network loss under an energy constraint, the authors develop a training framework including a novel greedy algorithm to compute the projection of the weight tensors to the energy constraint.

Pros:

The proposed method allows to accurately impose an energy constraint (in terms of the proposed model), in contrast to previous methods, and also yields a higher accuracy than these on some data sets. The proposed solution seems sound (although I did not check the proofs in detail, and I am not very familiar with hardware energy consumption subtleties).

Questions:

The experiments in Sec. 6.2 suggest that the activation mask is mainly beneficial when the data is highly structured. How are the benefits (in terms of weight and activation sparsity) composed in the experiments on Imagenet? How does the weight sparsity of the the proposed method compare to the related methods in these experiments? Is weight sparsity in these cases a good proxy for energy consumption?

How does the activation sparsity (decay) parameter (\delta) q affect the accuracy-energy consumption tradeoff for the two data sets?

The authors show that the weight projection problem can be solved efficiently. How does the guarantee translate into wall-clock time?

Filter pruning methods [1,2] reduce both the size of the weight and activation tensors, while not requiring to solve a complicated projection problem or introducing activation masks. It would be good to compare to these methods, or at least comment on the gains to be expected under the proposed energy consumption model.

Knowledge distillation has previously been observed to be quite helpful when constraining neural network weights to be quantized and/or sparse, see [3,4,5]. It might be worth mentioning this.

Minor comments:
- Sec. 3.4. 1st paragraph: subscript -> superscript
- Sec. 6.2 first paragraph: pattens -> patterns, aliened -> aligned

[1] He, Y., Zhang, X., & Sun, J. (2017). Channel pruning for accelerating very deep neural networks. ICCV 2017.
[2] Li, H., Kadav, A., Durdanovic, I., Samet, H., & Graf, H. P. Pruning filters for efficient convnets. ICLR 2017.
[3] Mishra, A., & Marr, D. Apprentice: Using knowledge distillation techniques to improve low-precision network accuracy. ICLR 2018.
[4] Tschannen, M., Khanna, A., & Anandkumar, A. StrassenNets: Deep learning with a multiplication budget. ICML 2018.
[5] Zhuang, B., Shen, C., Tan, M., Liu, L., & Reid, I. Towards effective low-bitwidth convolutional neural networks. CVPR 2018.

---

> ### Author Response · Authors · 2018-11-13
> **Responding to your comments (part 2)**
>
> > “The authors show that the weight projection problem can be solved efficiently. How does the guarantee translate into wall-clock time?”
>
> The most time consuming part of our proposed projection method is sorting the “profit density” in Algorithm 2. This sorting takes O(n logn) theoretical time complexity (n is the number of weights in DNN), and can be efficiently computed on GPUs using dedicated CUDA libraries.
> We measured the wall-clock time of our projection algorithm on a GPU server (CPU: Xeon E3 1231-v3, GPU: GTX 1080 Ti), and the result is (the time is averaged over 100 iterations):
> +-------------------------------------+------------+------------------+--------------------+
> |            DNNs                          | AlexNet | SqueezeNet | MobileNetV2 |
> +-------------------------------------+------------+------------------+--------------------+
> | Projection Time (seconds) |  0.170    |    0.023           |    0.032            |
> +-------------------------------------+------------+------------------+--------------------+
> As the data shows, the projection step can be solved very efficiently. We will include these results in the revision.
>
>
>
> > “Filter pruning methods [1,2] reduce both the size of the weight and activation tensors, while not requiring to solve a complicated projection problem or introducing activation masks. It would be good to compare to these methods, or at least comment on the gains to be expected under the proposed energy consumption model.”
>
> Filter pruning methods [1,2] require a sparsity ratio to be set for each layer, and these sparsity hyper-parameters will determine the energy cost of the DNN. Manually setting all these hyper-parameters in energy constrained compression is not trivial. NetAdapt [Yang et al., 2018] proposes a heuristic-driven approach to search such sparsity ratios and use filter pruning as proposed in [2] to train DNN models. In the paper, we directly compared against NetAdapt, and show that we can achieve higher accuracy with lower/same energy consumption. Please see Table 1 and Table 2.
>
>
> > “Knowledge distillation has previously been observed to be quite helpful when constraining neural network weights to be quantized and/or sparse, see [3,4,5]. It might be worth mentioning this.”
>
> Thank you for pointing out this point. We did notice several recently papers that use knowledge distillation for quantization and compression, and we will emphasize this with the suggested references in the revision.

---

> > ### Comment · AnonReviewer3 · 2018-11-29
> > **Response to the rebuttal**
> >
> > Thank you for the detailed response. I think the paper became more convincing and I will adapt my rating.

---

> ### Author Response · Authors · 2018-11-13
> **Responding to your comments (part 1)**
>
> Thanks for your thoughtful comments. The posted questions are answered as follows.
>
> > “The experiments in Sec. 6.2 suggest that the activation mask is mainly beneficial when the data is highly structured. How are the benefits (in terms of weight and activation sparsity) composed in the experiments on Imagenet? How does the weight sparsity of the the proposed method compare to the related methods in these experiments? Is weight sparsity in these cases a good proxy for energy consumption?”
>
> As the reviewers pointed out, the activation mask applies to cases where the data is highly structured. It does not apply to data from ImageNet. We acknowledge at the beginning of Section 3.2 that “We do not claim that applying input mask is a general technique; rather, we demonstrate its effectiveness when applicable.”
>
> In this work sparsity is not the end goal. Rather, it is a byproduct of energy saving. In fact, we observe that weight spartisy is *not* a good proxy for energy consumption, as also confirmed by prior work EAP [Yang et al., CVPR 2017]. Our method achieves lower energy consumption despite having higher density. The sparsity result on ImageNet is shown as follows. We will add the results in the revision.
> +-------------------------+--------------------------------+---------------------------------+------------------------+
> |       DNNs               |        AlexNet                    |       SqueezeNet               | MobileNetV2       |
> +-------------------------+--------------------------------+---------------------------------+------------------------+
> | Methods                | MP | SSL | EAP | Ours | MP  | SSL | EAP | Ours | MP  | SSL | Ours|
> +-------------------------+------+-------+------+--------+-------+-------+------+--------+-------+-------+-------+
> | Weights Sparsity  | 8% | 35% |  9% |  31% | 34% | 61%| 28%|  48% | 52% | 63%|  63%|
> +-------------------------+------+-------+------+--------+-------+-------+------+------+----------+------+-------+
>
>
> > “How does the activation sparsity (decay) parameter (\delta) q affect the accuracy-energy consumption tradeoff for the two data sets?”
>
> The decay parameter $\delta q$ is used to make the tradeoff between training time and accuracy. Smaller $\delta q$ leads to better accuracy, however, we need to run more outer loops of Algorithm 1. As shown in Algorithm 1, the outer loop is time consuming since it requires training of both W and M. Although smaller $\delta q$ could improve the accuracy of our method, we simply set $\delta q = 0.1|M|$ in all the experiments.

---

### Official Review · AnonReviewer2 · 2018-11-02
**The good paper, there are several questions**

**Rating:** 7
**Confidence:** 3

**Review:**

The paper is dedicated to energy-based compression of deep neural networks. While most works on compression are dedicated to decreasing the number of parameters or decreasing the number of operations to speed-up or reducing of memory footprint, these approaches do not provide any guarantees on energy consumption. In this work the authors derived a loss for training NN with energy constraints and provided an optimization algorithm for it. The authors showed that the proposed method achieves higher accuracy with lower energy consumption given the same energy budget. The experimental results are quite interesting and include even highly optimized network MobileNetV2.

Several questions and concerns.
‘Our energy modeling results are validated against the industry-strength DNN hardware simulator ScaleSim’. Could the authors please elaborate on this sentence?

One of the main assumptions is the following. If the value of the data is zero, the hardware can skip accessing the data. As far as I know, this is a quite strong assumption, that is not supported by many architectures. How do the authors take into account overhead of using sparse data formats in such hardware in their estimations? Is it possible to simulate such behavior in ScaleSim? Moreover, in many modern systems DRAM can only be read in chunks. Therefore it can decrease number of DRAM accesses in (4).

Small typos and other issues:
Page 8. ‘There exists an algorithm that can find an an \epsilon’
Page 8.’ But it is possible to fan approximate solution’
Page 4.  It is better to put the sentence ‘where s convolutional stride’  after (2).
In formulation of the Theorem 3, it is better to explicitly state that A contains rational numbers only since gcd is used.
Overall, the paper is written clearly and organized well, contains interesting experimental and theoretical results.

---

> ### Author Response · Authors · 2018-11-13
> **Responding to your comments**
>
> Thanks for your comments on our paper.
>
> > “‘Our energy modeling results are validated against the industry-strength DNN hardware simulator ScaleSim’. Could the authors please elaborate on this sentence?”
>
> ScaleSim simulates the DNN hardware execution cycle by cycle, from which it derives the total execution time and energy consumption of executing a network on the hardware. In this paper, we model the energy consumption of an network analytically (Section 3, in particular Equation 16); we compare the energy consumption analytical derived by our approach with the energy consumption estimated from ScaleSim (which simulates the hardware executions), and found that the two matched.
>
>
> > “One of the main assumptions is the following. If the value of the data is zero, the hardware can skip accessing the data. As far as I know, this is a quite strong assumption, that is not supported by many architectures. How do the authors take into account overhead of using sparse data formats in such hardware in their estimations? Is it possible to simulate such behavior in ScaleSim? Moreover, in many modern systems DRAM can only be read in chunks. Therefore it can decrease number of DRAM accesses in (4).”
>
> In many today’s DNN hardware the activations and weights are compressed in the dense form, and thus only non-zero values will be accessed. This is done in prior work [Chen et al., 2016; Parashar et al., 2017]. There is a negligible amount of overhead to “unpack” and “pack” compressed data, which we simply take away from the energy budget as a constant factor. This is also the same modeling assumption used by EAP [Yang et al., CVPR 2017].
>
> We agree with the reviewer that DRAM is accessed in bursts, which we did account for in our modeling. In particular, the per-access energy eDRAM we used in the modeling is the amortized energy of each access across the entire bursts. That is, instead of decreasing the number of DRAM accesses, we decrease the per-access energy. This is a standard modeling assumption widely used in the hardware architecture community and industry [Han et al, ISCA 2016; Yang et al., CVPR 2017].

---

> > ### Comment · AnonReviewer2 · 2018-11-25
> > **final comment**
> >
> > I would like to thanks the authors for the response that clarifies my questions. I would suggest adding several lines describing the overhead of packing and unpacking of sparse representation in the final revision of the paper. I agree with the authors that methods from Louizos et al., NIPS'17 and Neklyudov et al., NIPS'17 are quite orthogonal to the method considered in the paper. Nevertheless, these methods are strong baselines and improving them is a good indicator of the significance of the proposed method of pruning input channels.

---

### Official Review · AnonReviewer1 · 2018-11-03
**Interesting idea for energy-constrained compression, but some improvements still possible**

**Rating:** 7
**Confidence:** 4

**Review:**

This paper describes a procedure for training neural networks via an explicit constraint on the energy budget, as opposed to pruning the model size as commonly done with standard compression methods.  Comparative results are shown on a few data sets where the proposed method outperforms multiple different approaches.  Overall, the concept is interesting and certainly could prove valuable in resource-constrained environments.  Still I retain some reservations as detailed below.

My first concern is that this paper exceeds the recommended 8 page limit for reasons that are seemingly quite unnecessary.  There are no large, essential figures/tables, and nearly the first 6 pages is just introduction and background material.  Likewise the paper consumes a considerable amount of space presenting technical results related to knapsack problems and various epsilon-accurate solutions, but this theoretical content seems somewhat irrelevant and distracting since it is not directly related to the greedy approximation strategy actually used for practical deployment.  Much of this material could have been moved to the supplementary so as to adhere to the 8 page soft limit.  Per the ICLR reviewer instructions, papers deemed unnecessarily long relative to this length should be judged more critically.

Another issue relates to the use of a mask for controlling the sparsity of network inputs.  Although not acknowledged, similar techniques are already used to prune the activations of deep networks for compression.  In particular, various forms of variational dropout essentially use multiplicative weights to remove the influence of activations and/or other network components similar to the mask M used is this work.  Representative examples include Neklyudov et al., "Structured Bayesian Pruning via Log-Normal Multiplicative Noise," NIPS 2017 and Louizos et al., "Bayesian Compression for Deep Learning," NIPS 2017, but there are many other related alternatives using some form of trainable gate or mask, possibly stochastic, to affect pruning (the major ML and CV conferences over the past year have numerous related compression papers).  So I don't consider this aspect of the paper to be new in any significant way.

Moreover, for the empirical comparisons it would be better to compare against state-of-the-art compression methods as opposed to just the stated MP and SSL methods from 2015 and 2016 respectively.  Despite claims to the contrary on page 9, I would not consider these to be state-of-the-art methods at this point.

Another comment I have regarding the experiments is that hyperparameters and the use of knowledge distillation were potentially tuned for the proposed method and then simultaneously applied to the competing algorithms for the sake of head-to-head comparison.  But to me, if these enhancements are to be included at all, tuning must be done carefully and independently for each algorithm.  Was this actually done?  Moreover it would have been nice to see results without the confounding influence of distillation to isolate sources of improvement, but no ablation studies were presented.

Finally, regarding the content in Section 5, the paper carefully presents an explicit bound on energy that ultimately leads to a constraint that is NP-hard just to project on to, although approximate solutions exist that depend on some error tolerance.  However, even this requires an algorithm that is dismissed as "complicated."  Instead a greedy alternative is derived in the Appendix which presumably serves as the final endorsed approach.  But at this point it is no longer clear to me exactly what performance guarantees remain with respect to the energy bound.  Theorem 3 presents a fairly inscrutable bound, and it is not at all transparent how to interpret this in any practical sense.  Note that after Theorem 3, conditions are described whereby an optimal projection can be obtained, but these seem highly nuanced, and unlikely to apply in most cases.

Additionally, it would appear that crude bounds on the energy could also be introduced by simply penalizing/constraining the sparsity on each layer, which leads to a much simpler projection step.  For example, a simple affine function of the L0 norm would be much easier to optimize and could serve as a loose bound on the energy, given that the latter should be a non-decreasing function of the L0 norm.  Any idea how such a bound compares to those presented given all the approximations and greedy steps that must be included?


Other comments:
- As an implementation heuristic, the proposed Algorithm 1 gradually decays the parameter q, which controls the sparsity of the mask M.  But this will certainly alter the energy budget, and I wonder how important it is to employ a complex energy constraint if minimization requires this type of heuristic.

- I did not see where the quantity L(M,W) embedded in eq. (17) was formally defined, although I can guess what it is.

- In general it is somewhat troublesome that, on top of a complex, non-convex deep network energy function, just the small subproblem required for projecting onto the energy constraint is NP-hard.  Even if approximations are possible, I wonder if this extra complexity is always worth it relative so simple sparsity-based compression methods which can be efficiently implemented with exactly closed-form projections.

- In Table 1, the proposed method is highlighted as having the smallest accuracy drop on SqueezeNet.  But this is not true, EAP is lower.  Likewise on AlexNet, NetAdapt has an equally optimal energy.

---

> ### Author Response · Authors · 2018-11-13
> **Responding to your comments (part 3)**
>
> > “As an implementation heuristic, the proposed Algorithm 1 gradually decays the parameter q, which controls the sparsity of the mask M.  But this will certainly alter the energy budget, and I wonder how important it is to employ a complex energy constraint if minimization requires this type of heuristic.”
>
> The purpose of our proposed energy constraint is to exactly characterize the dependence between the sparsity of all parameters and the energy consumption, which provides us an (almost) exact energy model and a clear goal to guide us to pursue an energy efficient model. However, due to the nontrivial structure in the energy model, we have to involve some heuristics to solve it approximately.
>
>
> > “I did not see where the quantity L(M,W) embedded in eq. (17) was formally defined, although I can guess what it is.”
>
> Thanks for pointing this out. L is the original loss, e.g., cross-entropy loss for classification. We will clarify this in the revision.
>
>
> > “In general it is somewhat troublesome that, on top of a complex, non-convex deep network energy function, just the small subproblem required for projecting onto the energy constraint is NP-hard.  Even if approximations are possible, I wonder if this extra complexity is always worth it relative so simple sparsity-based compression methods which can be efficiently implemented with exactly closed-form projections.”
>
> Although the energy constrained problem is complex, our main contribution is to simplify it and propose an efficient method to solve it approximately. We measure the wall-clock time of the projection step, and across AlexNet, Squeezenet, and MobileNetV2, the projection step can be solved extremely efficiently -- within 0.2 seconds to be exact. Please also see our response to the 3rd question from Reviewer 3.
>
> In addition, at the technique-level, using a simple sparsity-based compression method to train energy-constrained DNNs would require setting the sparsity threshold for each layer to satisfy the energy constraint while minimizing the loss. Such a hyper-parameter tuning is not trivial. We compare against one such method (NetAdapt) and demonstrate higher accuracy with lower/same energy (Please see Table 1 and Table 2).
>
>
> > “In Table 1, the proposed method is highlighted as having the smallest accuracy drop on SqueezeNet.  But this is not true, EAP is lower.  Likewise on AlexNet, NetAdapt has an equally optimal energy.”
>
> In Table 1, our evaluation methodology is to configure our method to have an energy that is *the same as or lower than the lowest energy of prior work*, and compare the accuracy drops. In the case of AlexNet, our approach has a lower accuracy drop compared to NetAdapt at the same energy consumption. In the case of SqueezeNet, we show that our approach has the lowest energy among all the methods with only 0.3% higher accuracy drop than EAP. In Figure 2, we perform a comprehensive study where we vary the energy consumption of our method. We show that our method can train a network that has lower energy and less accuracy drop (the rightmost solid blue square) compared to EAP.
>
> We will clarify our writing in the revision.

---

> > ### Comment · AnonReviewer1 · 2018-11-29
> > **thanks for the clarifications**
> >
> > The authors provided reasonable clarifications, so I will bump up my score.

---

> ### Author Response · Authors · 2018-11-13
> **Responding to your comments (part 2)**
>
> > “Another comment I have regarding the experiments is that hyperparameters and the use of knowledge distillation were potentially tuned for the proposed method and then simultaneously applied to the competing algorithms for the sake of head-to-head comparison.  But to me, if these enhancements are to be included at all, tuning must be done carefully and independently for each algorithm.  Was this actually done?  Moreover it would have been nice to see results without the confounding influence of distillation to isolate sources of improvement, but no ablation studies were presented.”
>
> In our early experiments, we did not use knowledge distillation in other methods and found that the performance is significantly worse than ours. Therefore, we apply the knowledge distillation in all the methods for fair comparison. Recent work (e.g., [Mishra et al., 2018] ) also support our observation that the knowledge distillation trick can significantly improve the test accuracy in other pruning methods. We verified that the performance was not very sensitive to the value of lambda as long as lambda is in a reasonable range. Therefore, we empirically choose lambda to be 0.5 universal to *all* datasets.
>
> [Mishra et al., 2018] Mishra, Asit, and Debbie Marr. "Apprentice: Using knowledge distillation techniques to improve low-precision network accuracy." In ICLR 2018.
>
>
> > “Finally, regarding the content in Section 5, the paper carefully presents an explicit bound on energy that ultimately leads to a constraint that is NP-hard just to project on to, although approximate solutions exist that depend on some error tolerance.  However, even this requires an algorithm that is dismissed as "complicated."  Instead a greedy alternative is derived in the Appendix which presumably serves as the final endorsed approach.  But at this point it is no longer clear to me exactly what performance guarantees remain with respect to the energy bound.  Theorem 3 presents a fairly inscrutable bound, and it is not at all transparent how to interpret this in any practical sense.  Note that after Theorem 3, conditions are described whereby an optimal projection can be obtained, but these seem highly nuanced, and unlikely to apply in most cases.”
>
> The conditions under Theorem 3 are sufficient conditions to obtain the exactly optimal projection, i.e. the error bound is 0. However, we usually do not require such rigorous result in practice. Because the amount of parameters is very large in DNNs, the remaining budget R(W’’) is usually very small compared to E_budget. Therefore, the projection error bound is small enough in most cases.
> Another practical aspect of Theorem 3 is quantifying the upper-bound of the projection error in (27). In practice, we can exactly calculate this error bound and even choose to use the more accurate (but slower) algorithm in Theorem 2 when this error bound is not acceptable.
>
>
> > “Additionally, it would appear that crude bounds on the energy could also be introduced by simply penalizing/constraining the sparsity on each layer, which leads to a much simpler projection step.  For example, a simple affine function of the L0 norm would be much easier to optimize and could serve as a loose bound on the energy, given that the latter should be a non-decreasing function of the L0 norm.  Any idea how such a bound compares to those presented given all the approximations and greedy steps that must be included?”
>
> To use a method based on sparsity constraint of each layer, one must identify the sparsity bound for each of the DNN layers in a way that the whole model satisfies the energy budget while minimizing the loss. Even if an affine function of a layer’s sparsity bound can be used to estimate the layer’s energy, we still need to optimize these sparsity variables collectively across all layers for the whole model. Thus, the effectiveness of the layer-wise approach rests upon if we could find the optimal sparsity combination for all the layers.
>
> NetAdapt [Yang et al., ECCV 2018] and AMC [He et al., ECCV 2018] already showed that it is non-trivial to find the optimal layer-wise spartisy bounds. NetAdapt proposed a heuristic-driven search algorithm. In our experiment, we compared against NetAdapt and show that we can achieve higher accuracy with lower or same energy consumption (Please see Table 1 and Table 2).

---

> ### Author Response · Authors · 2018-11-13
> **Responding to your comments (part 1)**
>
> We very much appreciate your careful review. We clarify the questions point by point below and plan to sort out some confusions in our revision to improve the clarity.
>
> > “My first concern is that this paper exceeds the recommended 8 page limit for reasons that are seemingly quite unnecessary.  There are no large, essential figures/tables, and nearly the first 6 pages is just introduction and background material.”
>
> We think you refer to the Section 3.
> In Section 3, we show how the energy of a DNN inference is analytical modelled. We want to include these details in the paper because they form the final energy constraint proposed in problem (18). In the revised version, we will take your suggestion to reduce the number of pages to 8 by condensing this section and moving the details of energy estimation into the Appendix.
>
>
> > “Likewise the paper consumes a considerable amount of space presenting technical results related to knapsack problems and various epsilon-accurate solutions, but this theoretical content seems somewhat irrelevant and distracting since it is not directly related to the greedy approximation strategy actually used for practical deployment.  Much of this material could have been moved to the supplementary so as to adhere to the 8 page soft limit.”
>
> Thanks for your suggestion how to reduce the length to 8 pages. Please allow us to clarify the logic of our theorems first. All three algorithms are related to how to solve the key projection step in (22). Theorem 1 shows the projection problem in (22) is NP hard to find the exact optimal solution in general, since it is equivalent to a 0/1 knapsack problem. Theorem 2 shows the optimal computational complexity to find an epsilon *approximate* solution by utilizing the structure of the projection problem. Theorem 3 shows the proposed greedy algorithm (weighted projection algorithm) can achieve a reasonable precision efficiently. We feel that these theorems are useful in that they help understand the difficulty and the complexity of solving (22). We will consider moving Theorem 2 to the supplement in the first priority to shrink the length of this paper.
>
>
> > “Prior work also uses a mask for controlling the sparsity of network inputs, such as "Structured Bayesian Pruning via Log-Normal Multiplicative Noise," NIPS 2017 and Louizos et al., "Bayesian Compression for Deep Learning," NIPS 2017. How do you compare with them?”
>
> We agree that there is prior work that uses a mask to prune the network activations (i.e., inputs). But we want to emphasize two key differences of our work. First, these two papers you mentioned use the mask (structure sparsity) to remove unnecessary channels; whereas our work uses the mask to filter unimportant elements within each channel, motivated by the observation that many areas in the input image do not really contribute to the recognition task such as the corners of the input image in the digital number recognition. These two mask techniques are orthogonal, and can even be combined.
>
> Second, our mask model is integrated with an energy model to let us train energy-constrained DNNs; whereas these two papers purely aim at reducing the network parameters to get speedup.
>
> We use their released code to train energy-constrained DNNs on the MNIST dataset, the results are below:
> +--------------------------------------+----------------------+----------+-----------------------------+
> | Method                                  | Accuracy Drop | Energy | Width of Each Layer |
> +--------------------------------------+----------------------+----------+-----------------------------+
> | [Louizos et al., NIPS'17]      | 2.2%                   | 26%      | 4-6-52-42                    |
> +--------------------------------------+----------------------+----------+-----------------------------+
> | [Neklyudov et al., NIPS'17] | 1.5%                   | 22%     | 3-10-23-28                  |
> +--------------------------------------+----------------------+----------+-----------------------------+
> Our method has 0.5% accuracy drop with 17% energy cost, better than the two approaches.
>
>
> > “Comparison against methods newer than MP and SSL.”
>
> In the experiment, we also compare against state-of-the-art pruning methods NetAdapt  [Yang et al., ECCV 2018] and EAP [Yang et al., CVPR 2017] and show favorable results (Please refer to Table 1 and Table 2). SSL and MP are classic pruning techniques that represent a class of methods that use sparsity as the constraint (regularization). Indeed, EAP is the refined version of SSL and MP.

---

### Meta-Review · Area_Chair1 · 2018-12-12
**Good paper on minimizing energy cost in neural networks.**

**Confidence:** 5
**Recommendation:** Accept (Poster)

**Metareview:**

All of the reviewers agree that this is a well-written paper with the novel perspective of minimizing energy consumption in neural networks, as opposed to maximizing sparsity, which does not always correlate with energy cost. There are a number of promised clarifications and additional results that have emerged from the discussion that should be put into the final draft. Namely, describing the overhead of converting from sparse to dense representations, adding the Imagenet sparsity results, and adding the time taken to run the projection step.